# Chemical composition of ambient PM$_{2.5}$ over China and relationship to precursor emissions during 2005–2012

Guannan Geng[1], Qiang Zhang[1], Dan Tong[1,2], Meng Li[1,2], Yixuan Zheng[1], Siwen Wang[2], and Kebin He[2]

[1]Ministry of Education Key Laboratory for Earth System Modeling, Department of Earth System Science, Tsinghua University, Beijing, China
[2]State Key Joint Laboratory of Environment Simulation and Pollution Control, School of Environment, Tsinghua University, Beijing, China

*Correspondence to*: Qiang Zhang (qiangzhang@tsinghua.edu.cn)

**Abstract.** In this work, we presented the characteristics of PM$_{2.5}$ chemical composition over China for the period of 2005–2012, by synthesis of in-situ measurement data collected from literatures and satellite-based estimates using aerosol optical depth (AOD) data and the GEOS-Chem chemical transport model. We revealed the spatiotemporal variations in PM$_{2.5}$ composition during 2005–2012 and investigated the driving forces behind the variations by examining the changes in precursor emissions using a bottom-up emission inventory. Both in-situ observations and satellite-based estimates identified that secondary inorganic aerosols (i.e., sulfate, nitrate, and ammonium, SNA) ranked the highest fraction in dust-free PM$_{2.5}$ concentrations, followed by organic matters (OM) and black carbon (BC). For instance, satellite-based estimates found that SNA, OM and BC contributed to 59%, 33% and 8% of national population-weighted mean dust-free PM$_{2.5}$ concentrations respectively during 2005–2012. National population-weighted mean PM$_{2.5}$ concentration increased from 63.9 μg/m$^3$ in 2005 to 75.2 μg/m$^3$ in 2007, and subsequently decreased to 66.9 μg/m$^3$ from 2007 to 2012. Variations in PM$_{2.5}$ concentrations are mainly driven by the decrease of sulfate and the increase of nitrate. Population-weighted mean sulfate concentration decreased by 2.4% per year during 2005–2012 (from 14.4 μg/m$^3$ to 12.9 μg/m$^3$), while population-weighted mean nitrate concentration increased by 3.4% per year during 2005–2012 (from 9.8 μg/m$^3$ to 12.2 μg/m$^3$), largely offsetting the decrease of sulfate concentrations. By examining the emission data from the MEIC emission inventory, we found that the changes in sulfate and nitrate concentrations were in line with the decrease in SO$_2$ emissions and the increase in NO$_x$ emissions during the same period. The desulfurization regulation in power plants enforced around 2005 was the primary contributor to the SO$_2$ emissions reduction since 2006. In contrast, growth of energy consumption and lack of control measures for NO$_x$ resulted in persistent increase in NO$_x$ emissions until the installation of denitrification devices on power plants late in 2011, which began to take effect in 2012. The results of this work indicate that the synchronized abatement of emissions for multi-pollutants are necessary for reducing ambient PM$_{2.5}$ concentrations over China.

# 1 Introduction

Fine particulate matter with aerodynamic diameters of less than 2.5 μm ($PM_{2.5}$) is composed of a complex mixture originating from multiple sources, including sulfate ($SO_4^{2-}$), nitrate ($NO_3^-$), ammonium ($NH_4^+$), black carbon (BC), organic carbon (OC), crustal elements, and water. $PM_{2.5}$ and its chemical composition can penetrate deeply into human lungs and cause adverse health effects, including increased cardiovascular and respiratory morbidity and all-cause mortality (Dockery et al., 1993; McDonnell et al., 2000; Pope et al., 2002). Some components (e.g., sulfate, OC and BC) have significant impacts on the global energy budget system and consequently contribute to climate change (Bond et al., 2013; IPCC, 2013). $PM_{2.5}$ can also trigger visibility degradation or cause extreme haze events.

Measurements of the historical trends in air quality represent the basis of health impact studies and control policy assessments. However, such datasets are usually limited in developing countries, including China. Indeed, before 2013, the Chinese national monitoring network did not report measurements of $PM_{2.5}$ or its chemical composition, and thus, it is difficult to elucidate the historical changes in aerosols and their driving forces across China. Air quality changes prior to 2013 have attracted substantial attention because China experienced rapid economic growth and urbanization during that time, along with increased energy demand. Additionally, the Chinese government has begun taking actions to mitigate emissions. Knowledge of the variations in the $PM_{2.5}$ chemical composition and their relationship to precursor emissions can be used to improve the design of future plans.

Previous studies focusing on historical $PM_{2.5}$ concentrations trend analysis over China typically depended on measurement data from several sites (He et al., 2001; Sun et al., 2015) or simulations using chemical transport models (CTMs) (Wang et al., 2013; Xing et al., 2015). Measurements from individual cities are insufficient to support a comprehensive national analysis or health impact studies because air quality issues are usually regional problems and require knowledge of many different elements. Full understanding of the pollution sources can be achieved in combination with CTMs (Wang et al., 2013; Xing et al., 2015). However, CTMs have limitations in $PM_{2.5}$ simulations over China, since many models have been originally developed in other regions that has different pollution levels compared to China. Application of these models in China might introduce problems including missing precursors and formation mechanisms of secondary organic aerosols (Baek et al., 2011) and the lack of heterogeneous reactions, which may lead to underestimations of sulfate in haze events (Wang et al., 2014; Zheng et al., 2015). Therefore, additional information is needed to alleviate these issues and improve the simulations of historical $PM_{2.5}$ chemical compositions.

Satellite remote sensing of atmospheric pollutants has been widely used to understand the spatial and temporal distributions of air pollutants in recent years (Martin, 2008; Streets et al., 2013). Satellite retrievals can fill in the gaps in ground observations because of their high spatial and temporal coverage. Satellite remote sensing provides the column densities of trace gases (e.g., $SO_2$ and $NO_2$) and parameters that are related to aerosol concentrations, such as the aerosol optical depth (AOD), which has been widely used to estimate surface $PM_{2.5}$ concentrations (Chu et al., 2003; Wang and Christopher, 2003; van Donkelaar et al., 2016). Different types of methods can be used to retrieve ground-level $PM_{2.5}$ concentrations from

satellite AOD data, including the use of CTMs to obtain conversion factors between PM$_{2.5}$ and AOD (Liu et al., 2004; van Donkelaar et al., 2010) and the use of statistical models (Hu et al., 2014; Zheng et al., 2016) or semi-empirical models (Lin et al., 2015; Zhang et al., 2015) to investigate the relationship among PM$_{2.5}$, AOD and other factors. Compared to statistical models and semi-empirical models, CTMs do not require ground measurements as input data and can also derive PM$_{2.5}$ composition concentrations (Philip et al., 2014), making them suitable for studies seeking to explore the historical PM$_{2.5}$ chemical composition prior to 2013 over China. Our previous work improved the method of retrieving PM$_{2.5}$ concentrations using CTMs over China (Geng et al., 2015), and the method is used in this study.

In this work, we first attempted to fill in the gaps in PM$_{2.5}$ chemical composition measurements during 2005–2012 over China by using satellite AOD and conversion factors derived from a CTM. We also collected ground measurements of PM$_{2.5}$ composition from literatures to support the satellite-based analysis. The satellite-based concentrations were evaluated against the collected ground measurements. Based on in situ measurements and satellite-derived datasets, we investigated the spatial and temporal variations of PM$_{2.5}$ composition over China and identified the dominant species involved in these variations. We further compared the chemical composition concentrations with precursor emissions to better understand their relationship and the driving forces behind the changes.

## 2 Methodology

### 2.1 Ground-based measurement data

As mentioned previously, national-scale PM$_{2.5}$ and chemical composition measurements are unavailable over China during the study period (2005–2012). Therefore, we collected PM$_{2.5}$ and chemical composition measurements from the literatures. The sources, site locations, sampling period, and other information relevant to the collected data are summarized in Table S1. In total, we collected 96, 46, 46, 44, 53 and 55 records for PM$_{2.5}$, SO$_4^{2-}$, NO$_3^-$, NH$_4^+$, OC and BC, respectively. Although spatio-temporal continuous observation data are unavailable, the collected measurements cover most of the eastern provinces, are randomly distributed in time, and are considered to be representative of our study time and region for model evaluation.

### 2.2 Satellite-derived PM$_{2.5}$ and chemical composition concentrations

The satellite-derived PM$_{2.5}$ concentration datasets used in this work were adopted from Geng et al. (2015). These data were calculated using satellite AOD and conversion factors between AOD and PM$_{2.5}$ simulated by a CTM, and the spatial resolution of the dataset is $0.1° \times 0.1°$. Following Philips et al. (2014), the satellite-derived chemical compositions of PM$_{2.5}$ at $0.1° \times 0.1°$ were estimated by applying composition-specific conversion factors to satellite AOD. The equations used for the PM$_{2.5}$ and composition calculations are:

$$PM_{2.5,satellite} = AOD_{satellite} \cdot \frac{PM_{2.5,CTM}}{AOD_{CTM}} \tag{1}$$

$$\text{Composition}_{satellite}^{k} = \text{AOD}_{satellite} \cdot \frac{\text{Composition}_{CTM}^{k}}{\text{AOD}_{CTM}} \qquad (2)$$

where subscript 'satellite' and 'CTM' represent data from satellite and model respectively; $k$ represents different chemical compositions, including $SO_4^{2-}$, $NO_3^{-}$, $NH_4^{+}$, BC, OM, dust and sea salt, in this study.

The AOD retrievals were provided by the Moderate Resolution Imaging Spectroradiometer (MODIS, Levy et al., 2007) and Multi-angle Imaging SpectroRadiometer (MISR, Kahn et al., 2007) instruments onboard the Terra satellite. Daily AOD data at $0.1° \times 0.1°$ from the two instruments were filtered against ground AOD measurements from the Aerosol Robotic Network (AERONET, Holben et al., 1998) and China Aerosol Remote Sensing NETwork (CARSNET, Che et al., 2009) before averaging them to reduce the uncertainties. We used the nested-grid GEOS-Chem model v9-01-02 (Bey et al., 2001; for more details, see the Support Information) over Southeast Asia and year-by-year emission inventory over China to simulate the conversion factors between $PM_{2.5}$ species and AOD. The nested-grid model has a spatial resolution of $0.5°$ lat $\times 0.667°$ lon (Chen et al., 2009), which was driven by assimilated GEOS-5 meteorology from the Goddard Earth Observing System (GEOS) at the NASA Global Modeling and Assimilation Office (GMAO; http://gmao.gsfc.nasa.gov/). A total of eight years of simulations were conducted with initialization on January 1, 2005 after a one-year spin-up simulation. The study time period (i.e., 2005–2012) could not be expanded because of the restrictions imposed by the GEOS-5 meteorology data. In addition, the aerosol vertical profiles from the space-based Cloud-Aerosol Lidar with Orthogonal Polarization (CALIOP, Winker et al., 2007) used to adjust the model simulations and corrections for incomplete sampling were also taken into account. More details are provided in Geng et al. (2015) and Philips et al. (2014).

The uncertainties associated with satellite-derived $PM_{2.5}$ compositions mainly relate to the satellite AOD retrievals, the modeled conversion factor between $PM_{2.5}$ and AOD, the modeled $PM_{2.5}$ composition fractions and incomplete sampling because of missing AOD data (Philip et al., 2014). Since the satellite AOD data were filtered against ground AOD measurements, the uncertainty of the AOD was restricted to the larger bound of $\pm 20\%$ or $\pm 0.1$. Vertical profiles from the CALIOP instrument were used to adjust the modeled profile; therefore, the uncertainties in the modeled $PM_{2.5}$/AOD ratio were estimated as their annual mean differences. The normalized mean bias between each modeled component and ground observations was used as the error of the simulated $PM_{2.5}$ composition fraction. Uncertainty resulting from incomplete sampling was estimated as the relative difference between the modeled full-time mean value and the value coincidently sampled using satellite data. The total error was the sum of the described errors in quadrature.

## 2.3 Bottom-up emission inventory

The bottom-up emission information used in this study was taken from the Multi-resolution emission inventory for China (MEIC, http://www.meicmodel.org/). The MEIC model, which was developed and is maintained by Tsinghua University, is a technology-based, dynamic process emission model. The MEIC inventory is an update of the bottom-up emission inventory developed by the same group (Zhang et al., 2009) and includes estimated emissions from ~700 anthropogenic

sources over China. The power sector was estimated using a unit-based dataset, and the spatial distribution of the power emission was significantly improved (Wang et al., 2012; Liu et al., 2015). The on-road emissions were calculated using estimated county-level vehicle population data combined with an emission factor model (Zheng et al., 2014). We also built a framework to speciate the non-methane volatile organic compounds (NMVOCs) and provided model-ready NMVOC emissions for major chemical mechanisms (Li et al., 2014). In this work, the year-by-year emissions of $SO_2$, $NO_x$, $NH_3$, CO, OC, BC and NMVOCs during 2005–2012 were taken from the MEIC inventory as the GEOS-Chem model input to simulate the conversion factors between $PM_{2.5}$ and AOD, and to support the emission driving force analysis.

In the bottom-up method, large uncertainties lie in the activity data, emission factors, removal efficiency and distribution of control technology, and spatial proxies (Zhang et al., 2009), which could affect the estimation of emission trends. To evaluate the estimated emission trends in the MEIC inventory, simulated tropospheric vertical $SO_2$ and $NO_2$ columns from the GEOS-Chem model were compared with satellite observations from the Ozone Monitoring Instrument (OMI, Levelt et al., 2006). The satellite $SO_2$ products were taken from Wang et al. (2015), which were improved based on the standard NASA products to reduce uncertainties. The satellite $NO_2$ product was the OMI standard product, OMNO2 (version 2.1) (Bucsela et al., 2013). More descriptions of the satellite $SO_2$ and $NO_2$ columns are provided in the Support Information.

## 3 Model Evaluation

### 3.1 Satellite-derived concentration

Figure 1 compares the satellite-derived $PM_{2.5}$ and chemical composition concentrations with ground measurement data collected from the literature. Satellite-based data were extracted and averaged according to the corresponding sample period and location before comparison. All symbols were colored according to their site locations. Three regions covering most of the anthropogenic sources over China were defined in this study as follows: Eastern China (ECN, 28–42°N, 110–123°E), the Pearl River Delta (PRD, 20–25°N, 110–117°E) and the Sichuan Basin (SCB, 27–33°N, 102–110°E). We found good agreement between the satellite-based and in situ $PM_{2.5}$ concentrations, with R = 0.72, slope = 0.80 and intercept = 10.18 $\mu g/m^3$. However, satellite-derived $PM_{2.5}$ concentrations were underestimated by a factor of two in some places outside the three regions (e.g. Liaoning, Inner Mongolia and Fujian province), which might affect the health impact estimates in these regions. The underestimation was mainly caused by the biases in modeled $PM_{2.5}$ concentrations, and the satellite-derived data improved compared to the model results (Figure S1).

The chemical compositions estimated in this study also had good consistency with ground measurements, with R values in the range of 0.65-0.75 for different species. The satellite-derived data had relatively small biases in $SO_4^{2-}$ and OM estimations but overestimated $NO_3^-$, $NH_4^+$ and BC over polluted regions. The overestimation of $NO_3^-$ and $NH_4^+$ was caused by overestimation of the $NO_3^-$ fractions by the GEOS-Chem model, which was a common problem encountered in previous model studies (Park et al., 2004; Zhang et al., 2012; Wang et al., 2013). The overestimation of satellite-derived BC

concentrations mainly occurred in the ECN region. In general, the satellite-derived concentrations were improved relative to the model simulations (Figure S1).

## 3.2 Modeled $SO_2$ and $NO_2$ columns

To examine the model's capability for capturing the trends in $PM_{2.5}$ precursors, we also compared the normalized inter-annual trends of model-simulated and satellite-observed tropospheric $SO_2$ and $NO_2$ column densities over China, as shown in Figure 2. Satellite-derived $SO_2$ and $NO_2$ relative trends were well captured by the model simulations, with slight overestimation of the growth rates in some years. ECN had the best performance for both $SO_2$ and $NO_2$, with only a small overestimation of the $NO_2$ growth rates during 2008–2009 and an underestimation of the $SO_2$ growth rate in 2007. The relative increases in the $NO_2$ columns were generally overestimated for PRD and SCB, indicating the possible overestimation of the $NO_x$ emissions growth rates. The negative values in the satellite $SO_2$ columns over PRD were mainly caused by the strong interference of ozone absorption in the retrieval process, especially over clean regions where the $SO_2$ concentrations were low. However, the satellite-observed $SO_2$ columns showed a decreasing trend in PRD, which was consistent with the modeled results. It is worth noting that the satellite retrieved $SO_2$ and $NO_2$ column densities have uncertainties in their trends because of the row anomaly issue happened to the CCD detectors in OMI instrument and the impact of changing aerosol loadings on the satellite retrievals, which might contribute to the discrepancies between modeled and satellite data.

## 4 Results and Discussion

### 4.1 Spatiotemporal variations in the $PM_{2.5}$ chemical composition

Figure 3 presents the mass concentrations of $PM_{2.5}$ species and their speciation in $PM_{2.5}$ in seven regions over China. Ground measurements obtained from the literature were classified into the seven regions, in companion with coincidently sampled satellite-based data. North, Central, Northwest and Southwest China had a high observed $PM_{2.5}$ level (above 100.0 μg/m$^3$), which were well reproduced by the satellite-based estimation. The observed secondary inorganic aerosols (sulfate, nitrate and ammonium [SNA]) concentration exceeded half of the dust-free $PM_{2.5}$ concentration in six regions over China, with large values in Central (49.8 μg/m$^3$) and Northwest (46.5 μg/m$^3$) China, indicating the significant contribution of secondary aerosols to $PM_{2.5}$ mass. The observed OM and BC fractions varied between 29–57% and 3–10% respectively, comparing to 27–40% and 9–14% from sampled satellite data. The main difference between observed and satellite-derived speciation occurred in Northeast China where few ground measurement were available. OM had a larger fraction in South (36%) and Northeast (57%) China, possibly due to larger biogenic emissions in these regions.

We also provided the population-weighted mean concentrations of satellite-derived $PM_{2.5}$ species for each region, which were estimated using population data taken from the LandScan Global Population database (ORNL, 2010, Bright et al.,

2011). North China ranked the highest $PM_{2.5}$ level (94.7 μg/m$^3$), followed by Central (88.8 μg/m$^3$) and Northwest (70.9 μg/m$^3$) China. SNA concentrations were most abundant in North (46.9 μg/m$^3$) and Central (43.3 μg/m$^3$) China, while their fractions in the dust-free $PM_{2.5}$ were highest in East (63%) and North (61%) China. The regional population-weighted mean $PM_{2.5}$ composition concentrations were quite different from the averaged observations, implying the possible biases when
using measurements data from several sites to represent the region's level.

Figure 4 summarizes the $PM_{2.5}$ speciation from ground measurements and corresponding satellite-based data for 20 major cities across China. Only observations covering more than one year were used. Dust and sea salt were also excluded in the comparison. In general, $PM_{2.5}$ speciation patterns were well captured by satellite-derived data in most cities. SNA were major components of $PM_{2.5}$ over Chinese cities, which contributed 46–70% of the dust-free $PM_{2.5}$ concentrations in the 20
cities, comparable to the observed fractions of 44–73%. The city with the largest SNA fraction in dust-free $PM_{2.5}$ was Qinghai Lake, from both observation (73%) and satellite-based estimation (70%). OM was also an important component in $PM_{2.5}$, with averaged fraction of 31% and 32 % from ground measurements and satellite data separately among the cities. Baotou has the largest observed fraction of OM in dust-free $PM_{2.5}$, which was underestimated by satellite-based data. The fractions of BC were overestimated by satellite-based data in many cities, which was caused by the overestimation of BC
concentration mentioned above.

Figure 5 shows the multi-year mean spatial distributions of the satellite-derived $PM_{2.5}$ concentrations over China overlaid with ground measurements with sample periods exceeding one year during the corresponding timeframe. The spatial patterns of $PM_{2.5}$ were well captured by the satellite-based data, with concentration enhancements of the anthropogenic pollution sources, including in the North China Plain, the Yangtze River Delta, the SCB and Hunan-Hubei province. The population-
weighted mean mass concentrations of $PM_{2.5}$ over China were in the range of 63.9–75.2 μg/m$^3$ during 2005–2012 and were nearly twice the Chinese national standard. China had an overall negative $PM_{2.5}$ trend of 0.3% per year during 2005–2012; a positive trend of 18.2% per year was estimated in 2005–2007, followed by a negative trend of 2.7% per year in 2008–2012. According to the inter-annual $PM_{2.5}$ changes over China shown in Figure 6a, the $PM_{2.5}$ level exhibited a general downward trend after the peak year (2007), although a slight rebound of the $PM_{2.5}$ concentration was found in 2011. The $PM_{2.5}$ trend
estimated in this work was consistent with those of previous studies (Boys et al., 2014; Ma et al., 2015).

The inter-annual changes in the satellite-derived $PM_{2.5}$ concentrations for ECN, PRD and SCB are presented in Figure 6b-6d. Regional disparities were found in the time series of $PM_{2.5}$ concentrations, especially at the inflection points beginning decreases or rebounds. ECN's trend was similar to the national trend, whereas in the PRD, a persistent negative trend of 2.6 μg/m$^3$ per year in $PM_{2.5}$ after 2007 was observed, with no rebound. $PM_{2.5}$ levels in the SCB peaked in 2006, which was one
year before the national peak; subsequently, it increased again beginning in 2008 until decreasing in 2012.

The spatial distributions and time series of SNA are shown in Figure 7. The spatial patterns of SNA in the eastern part of China were similar to those of $PM_{2.5}$. Satellite-based data could well reproduce the regional heterogeneity of SNA, although $NO_3^-$ was overestimated over ECN. The sulfate concentrations over eastern China generally exceeded 15.0 μg/m$^3$, and the

highest values occurred in the SCB ($\geq$ 30 μg/m$^3$). Regarding the nitrate concentrations, the highest values were found in the North China Plain, reflecting the different emission characteristics of sulfate and nitrate. SNA showed quite different temporal variations relative to the total PM$_{2.5}$. The population-weighted mean sulfate concentrations over China showed a negative trend of 2.4% per year during the study period (from 14.4 μg/m$^3$ in 2005 to 12.9 μg/m$^3$ in 2012), as shown in Figure

7j, with a peak value in 2006 (15.9 μg/m$^3$) and a rebound in 2011 (Figure 6e). In contrast to $SO_4^{2-}$, the nitrate concentrations over China had an increasing trend of 3.4% per year (from 9.8 μg/m$^3$ in 2005 to 12.2 μg/m$^3$ in 2012), which partially compensated for the reduction in sulfate and caused the total PM$_{2.5}$ to drop one year after sulfate. In 2012, nitrate decreased for the first time, and sulfate decreased again, causing the total PM$_{2.5}$ to decline as well. The temporal trends of sulfate and nitrate also differed among the three regions. ECN and SCB had $SO_4^{2-}$ trends similar to the national trend, whereas in the

PRD, the $SO_4^{2-}$ concentrations decreased consistently after 2007. The SCB had a larger $NO_3^-$ growth rate compared to other regions, which could explain the earlier rebound of PM$_{2.5}$ in this region.

Figure 8 presents the spatial and temporal changes of carbonaceous aerosols. Satellite-based data performed well for estimating OM concentrations but tended to overestimate the BC concentrations in polluted regions, which was caused by the overestimation of modeled BC fractions by the GEOS-Chem model. The estimated OM concentrations were highlighted

in densely populated regions because organic carbon is mainly produced by residential combustion. BC hotspots were most apparent in populous regions and large cities, consistent with previous ground observation studies (Cao et al., 2007). Negative trends of 0.9% and 0.1% per year were found for OM and BC, respectively, and these trends held nearly steady over this period.

The spatial and temporal changes in mineral dust and sea salt are presented in Figure 9. High values of mineral dust

(exceeding 80.0 μg/m$^3$) were found over the desert regions (e.g., the Taklimakan Desert), which were the largest contributors to PM$_{2.5}$ in northwestern China. Sea salt was mainly distributed along the coast, and its concentrations were less than 1.0 μg/m$^3$. Dust and sea salt were mainly produced from natural sources and fluctuated during this period.

Based on the analysis described above, we found that sulfate and nitrate were the two dominant components driving the variations in the PM$_{2.5}$ concentration during 2005–2012, which is consistent with previous studies using CTMs (Xing et al.,

2015). Decreases in $SO_4^{2-}$ caused the total PM$_{2.5}$ concentrations to decline beginning in 2007, whereas $NO_3^-$ was the main reason for the PM$_{2.5}$ rebound observed in 2011. We also compared the relative abundances of nitrate and sulfate (i.e., $NO_3^-/SO_4^{2-}$) in different years to confirm the variations of these two species. The $NO_3^-/SO_4^{2-}$ ratio over China was generally less than 2, lower than the values reported for developed cities (e.g., 2–5 in Los Angeles, Kim et al., 2000). The $NO_3^-/SO_4^{2-}$ ratios were larger in the eastern part of China, because eastern part of China had stricter emissions standards and higher

vehicle populations (Zheng et al., 2014), and in western China (e.g., Sichuan, Chongqing, and Ningxia), where coal with higher sulfur contents is burned (Tang et al., 2008), resulting in higher emission factors of SO$_2$. During 2005–2012, the $NO_3^-/SO_4^{2-}$ ratios over China exhibited an increasing trend, further supporting the changes in the relative abundances of

sulfate and nitrate in the atmosphere and the distinct process of controlling $SO_2$ and $NO_x$ emissions in China. Thus, the contribution of the mobile sector to $PM_{2.5}$ increased, while the contribution from point sources decreased. Ground measurements also reflected the increasing trend in the $NO_3^-/SO_4^{2-}$ ratio over China. For example, Fu et al. (2014) found that the $NO_3^-/SO_4^{2-}$ ratio in Guangzhou increased from 0.31 to 0.69 during 2007–2011. Additionally, Tan et al. (2016) observed an

increase in the $NO_3^-/SO_4^{2-}$ ratio from 0.73 to 0.92 in Foshan during 2008–2012.

## 4.2 Relationship between the $PM_{2.5}$ composition and precursor emissions

The sulfate and nitrate concentrations were compared with the precursor emissions of $SO_2$ and $NO_x$ to elucidate the driving forces underlying the changes in the $PM_{2.5}$ components, as shown in Figure 10. We used the percent changes relative to the 2005 benchmark for the direct comparison of the concentrations and emissions. The relative changes in the $SO_2$ and $NO_x$

emissions were consistent with the changes in the sulfate and nitrate concentrations, respectively, although discrepancies were noted in the growth and reduction rates.

On the national scale, the relative changes in $SO_4^{2-}$ from 2005 to 2012 corresponded almost linearly to the relative changes in $SO_2$ emissions, whereas the relative increase in $NO_3^-$ during 2005–2012 was less than that of $NO_x$ emissions. During this time period, $NH_3$ emissions remained steady, and the magnitude was less than the sum of the $SO_2$ and $NO_x$ emissions on a

molecular basis, indicating that $NH_3$-limited conditions existed in China. Although the decrease of $SO_4^{2-}$ released free $NH_3$ in a 1:2 ratio, the amount of $NH_3$ released could not fully neutralize the increased nitric acid because the increase in the $NO_x$ emissions was more than twice the decrease in the $SO_2$ emissions (e.g., an 11% reduction in $SO_2$ emissions vs. a 48% increase in $NO_x$ emissions in 2012). This explains why the relative increase of $NO_3^-$ was less than that of $NO_x$ emissions (e.g., a 24% increase of $NO_3^-$ in 2012).

Among the three regions, the PRD had the largest reduction rates of $SO_2$ emissions and the lowest growth rates of $NO_x$ emissions, consistent with the faster decline of $SO_4^{2-}$ concentrations and slower increase of $NO_3^-$ concentrations in this region. The SCB had a larger growth rate of $NO_x$ emissions compared to other regions, which could explain the rapid increase of $NO_3^-$ concentrations. Differences were observed in the relative change rates of $SO_4^{2-}$ concentrations and $SO_2$ emissions. Possible explanations for these differences include uncertainties in the bottom-up $SO_2$ emission trend, as described in Sect.

3.2, and the influences of the varying meteorological conditions over the years. Similar to the national trend, the growth of the $NO_3^-$ concentration was not only affected by the $NO_x$ emission increase but also by changes in the $SO_4^{2-}$ concentrations via the thermodynamic equilibrium of SNA.

Figure 11 presents the inter-annual emission trends of $SO_2$ and $NO_x$ in four sectors taken from the MEIC inventory. The emissions trends estimated by the MEIC model are consistent with other studies (Lu et al., 2010; Lu et al., 2011; Zhao et al.,

2013; Zhao et al., 2013). We also established a hypothetical scenario of emission trends that assumed that the emission

factors and technology distributions remained unchanged during 2005–2012 to separate the effects of activity (A) and emission factor (EF) changes. The hypothetical scenario represented the emission changes caused by A changes alone, and the differences between the hypothetical scenario and the actual emissions reflected the emission changes caused by EF variations.

As shown in Figure 11, the total $SO_2$ emissions peaked in 2006 (34.3 Tg) and subsequently decreased. This reduction was mainly driven by the power sector, which is a major source of $SO_2$ emissions over China. China set a target to reduce national $SO_2$ emissions by 10% in the 11th Five Year Plan (FYP, 2006–2010) and took corresponding actions, including the mandatory installation of flue-gas desulfurization (FGD) in coal-fired power plants and the optimization of generation unit fleets by promoting large power plants and decommissioning small plants (The State Council of the People's Republic of

China 2006). Although the increasing electricity demand tended to increase $SO_2$ emissions by 68% during 2005–2011, the reduced $SO_2$ EFs completely reversed the $SO_2$ emissions in the power sector, causing them to decrease at a rate of 1.4 Tg/yr and confirming the success of FGD operation and the optimization of the mix of generation unit fleets. After reaching a bottom value of 28.4 Tg in 2010, the $SO_2$ emissions subsequently increased. This rebound was mainly caused by the growing activity rates in the industrial sector and limited control measures in iron and steel production. Therefore, successful control

of $SO_2$ emissions in the power sector was the key factor relating to $SO_4^{2-}$ reduction after 2007, whereas limited controls in the industrial sector contributed to the $SO_4^{2-}$ rebound.

The total $NO_x$ emissions over China showed completely different trends compared to $SO_2$ emissions because of the distinct control processes implemented for $SO_2$ and $NO_x$ by the Chinese government. China established a goal to reduce $NO_x$ emissions by 10% late in the 12th FYP (2011–2015) and pursued the installation of selective catalytic reduction (SCR)

equipment at power plants beginning in 2011. Previously, the control measures on $NO_x$ in the power sector were limited, whereas the energy demand increased continuously, leading to a dramatic increase in $NO_x$ emissions from the power sector during 2005–2011. Meanwhile, the Chinese government promoted the construction of precalciner kilns, which are the most energy-efficient cement kilns, to replace the shaft kilns for cement production. However, the $NO_x$ EF increased because the higher operational temperatures and more automated air-flow systems of precalciner kilns resulted in higher $NO_x$ emissions

compared to shaft kilns. In the transport sector, $NO_x$ emissions increased by 19.3% in 2012 relative to 2005; this value was the result of a 17.4% decline because of the implementation of staged regulations and a 36.7% increase attributed to vehicle population growth. In comparison to 2011, the $NO_x$ emissions reduction caused by a decline in the activity level in the power sector in 2012 was estimated to be 26.5 Gg, much less than the estimated emission reduction of 897.5 Gg, implying that SCR promotion might have begun to take effect. The reduction of $NO_x$ emissions have also been noticed by other studies using

satellite retrievals (de Foy et al., 2016; Liu et al., 2016; van Der A et al., 2016). The rapid increase of $NO_x$ emissions caused the $NO_3^-$ concentrations to rise, compensating for the reductions in the $SO_4^{2-}$ concentration and even leading to a rebound in the $PM_{2.5}$ concentration.

Figure 12 summarizes the region-specific by-sector emission change rates caused by A and EF changes, which could facilitate understanding the regional disparities in PM$_{2.5}$ composition variations. ECN exhibited emission patterns that were very similar to those observed at the national level, and the SO$_2$ control measures implemented in the power sector were most effective in ECN. Indeed, in this region, a reduction rate of 17.4% per year was attributed to the EF reduction, larger than those observed in the PRD (11.4%/year) and SCB (9.9%/year).

The PRD had more effective control of power sector NO$_x$ emissions, and the vehicle population in this region grew more slowly, contributing to slower growth of the total NO$_x$ emission and, thus, persistent PM$_{2.5}$ reduction after 2007. In the SCB, coal consumption in the industrial sector grew quickly, and as a result, the SO$_2$ and NO$_x$ emissions increased by 20.9% and 20.0% per year, respectively. Reductions in the sulfur content of the coal burned in the SCB reduced the SO$_2$ EFs and offset a large fraction of the increased emissions. In contrast, for NO$_x$ emissions, the EFs continued to increase, causing the NO$_x$ emissions and NO$_3^-$ concentrations over the SCB to increase dramatically.

## 4.3 Uncertainties and limitations

The national population-weighted mean uncertainties were estimated to be 3.4 µg/m$^3$ for sulfate, 2.7 µg/m$^3$ for nitrate, 2.0 µg/m$^3$ for ammonium, 1.2 µg/m$^3$ for BC and 4.6 µg/m$^3$ for OM. Despite the large uncertainties in the magnitudes of the estimated PM$_{2.5}$ compositions, the trend analysis performed here was relatively unaffected because most of the uncertainties were systematic errors, and we used the normalized trend to cancel out most of them. The bottom-up emissions used to elucidate the driving forces were also expected to include large uncertainties, but we focused on the emission trend, which could be more reliable. In addition, we used a CTM and satellite observations from the OMI instrument to further support the estimated precursor emissions, which increased the robustness of our study.

Another limitation of our study is that the influence of meteorological conditions was not considered. Indeed, meteorological parameters, such as temperature, humidity and precipitation, influence the gas-phase formation of sulfate, gas-to-aerosol partitioning, and the wet and dry depositions of aerosols, which all contribute to the inter-annual variations in the PM$_{2.5}$ concentrations. According to Mu and Liao (2014), PM$_{2.5}$ concentrations exhibit large inter-annual variations in North China, especially during winter, when the meteorological parameters vary greatly. Indeed, North China had larger inter-annual variations in summer, whereas the SCB exhibited the smallest inter-annual variations among the polluted regions. In this work, we used annual mean values to compare concentrations and emissions, which might have partially reduced the variation of the uncertainties between seasons. However, future work is needed to more accurately quantify the relationship between precursor emissions and aerosol concentrations.

## 5 Concluding Remarks

In this study, we investigated the spatiotemporal variations of PM$_{2.5}$ compositions over China during 2005–2012, a period for which national ground measurements are unavailable, based on ground measurements obtained from literature and satellite-

derived datasets, and sought to identify the driving forces underlying these changes. We found that SNA ranked the highest fraction in dust-free $PM_{2.5}$ concentrations (52–63% in different regions), followed by OM and BC, which accounted for 29–39% and 7–10% of $PM_{2.5}$ respectively. The estimated national population-weighted mean $PM_{2.5}$ concentration increased from 2005 to 2007, and subsequently decreased from 2007 to 2012. Of the three polluted regions studied here, the PRD was

the only one to exhibit a persistent decrease in $PM_{2.5}$ after 2007; other regions had different peak and rebound years. The decline in the total $PM_{2.5}$ concentrations after 2007 was mainly attributable to sulfate reduction, whereas the rebound in $PM_{2.5}$ in 2011 was caused by persistent nitrate growth and a rebound in the sulfate concentration. The ratios between nitrate and sulfate increased over China from 2005 to 2012, further confirming the relative abundance changes of these two species. Bottom-up emission inventories of the precursors $SO_2$ and $NO_x$ were examined to explain the variations in these species.

FGD operation in the power sector was the primary contributor to $SO_2$ reduction, whereas the growth of industrial emissions caused $SO_2$ emissions to rebound in 2011. Limited control measures implemented during 2005–2011 caused the $NO_x$ emissions to increase rapidly, whereas the SCR system installed in the power sector during the 12th FYP caused the $NO_x$ emissions to decline in 2012. The $SO_2$ and $NO_x$ emission trends were generally consistent with the sulfate and nitrate concentrations, respectively, but exhibited different relative change rates, which could be explained by the thermodynamic

equilibrium of SNA.

In this study, our analysis mainly focused on the three selected regions (i.e., ECN, SCB and PRD), because they have high $PM_{2.5}$ levels, large anthropogenic emissions and large population densities. However, other places outside the three regions also experienced rapid industrialization and urbanization in recent years due to the national development strategies, which is important but has been paid less attention to. Future works could further improve the estimation of $PM_{2.5}$ composition

datasets in these area and investigate the spatiotemporal variations of $PM_{2.5}$ concentrations and its driving forces.

This study's findings show that the simultaneous regulation of $SO_2$ and $NO_x$ is crucial for $PM_{2.5}$ mitigation in China. A recent study (Cheng et al., 2016) reported that $NO_x$ is not only a precursor for nitrate but is also an important oxidant contributing to sulfate formation in northern China. This finding highlights the importance of controlling $NO_x$ emissions. The government has already implemented stricter control measures to reduce $SO_2$ and $NO_x$ emissions in recent years. FGD

devices have been promoted in iron and steel production to reduce industrial $SO_2$ emissions, and the use of SCR devices in the power sector also resulted in significant effects, causing the trend in $NO_x$ emissions to decline since 2011 (de Foy et al., 2016; Liu et al., 2016; van Der A et al., 2016). Ground measurements from the national monitoring network established in 2013 demonstrate a decreasing trend in $PM_{2.5}$ over China since 2013, confirming the efficacy of the emissions controls implemented in China. In the future, more attention should be paid to curbing $NO_x$ emissions.

**Competing interests**

The authors declare that they have no conflict of interest.

## Acknowledgements

This work was supported by the National Natural Science Foundation of China (41625020 and 41571130032) and China's National Key R&D Research Program (2016YFC0201506 and 2016YFC0208801). We thank the two anonymous reviewers for their constructive comments.

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

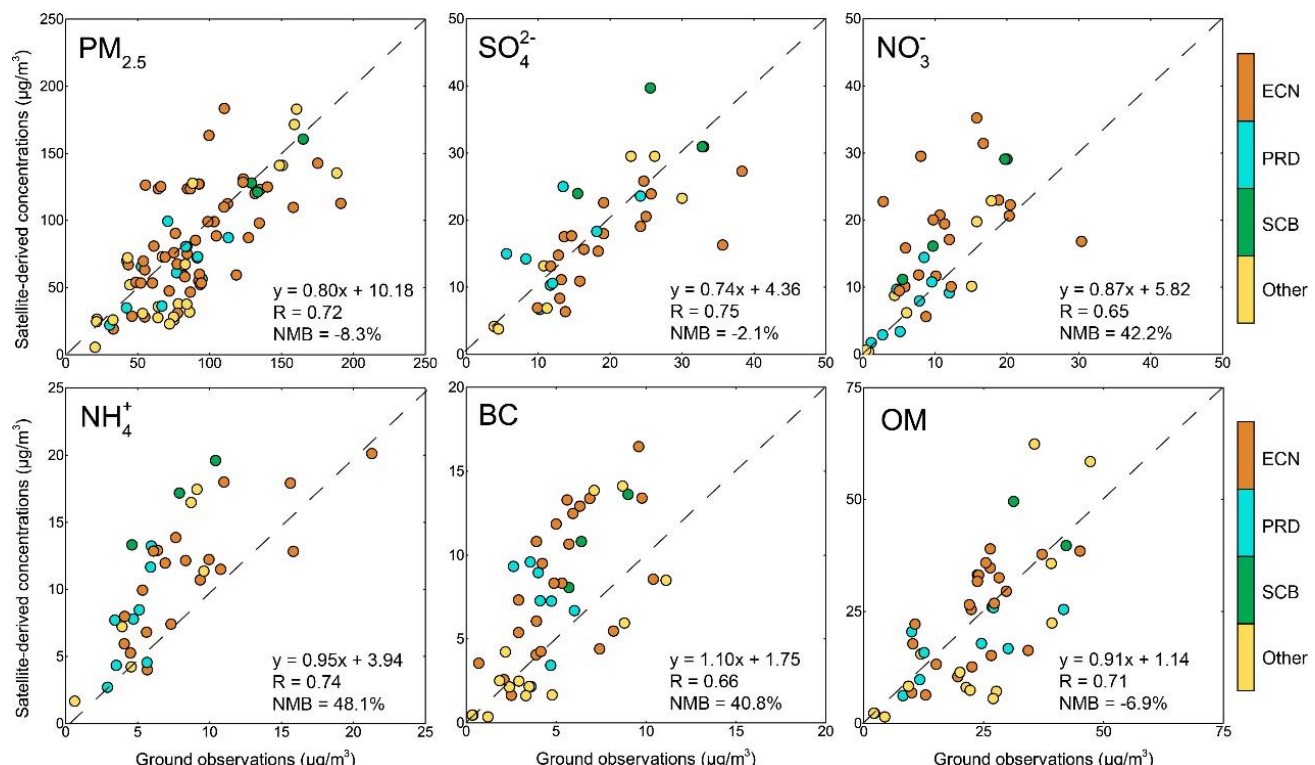

**Figure 1. Evaluation of satellite-derived PM$_{2.5}$ and chemical composition concentrations using in situ data collected from the literature. The colors of the symbols represent the locations of the collected in situ data. The three defined regions are ECN (28–42°N, 110–123°E), PRD (20–25°N, 110–117°E) and SCB (27–33°N, 102–110°E). The dashed line corresponds to the 1:1 line in each panel.**

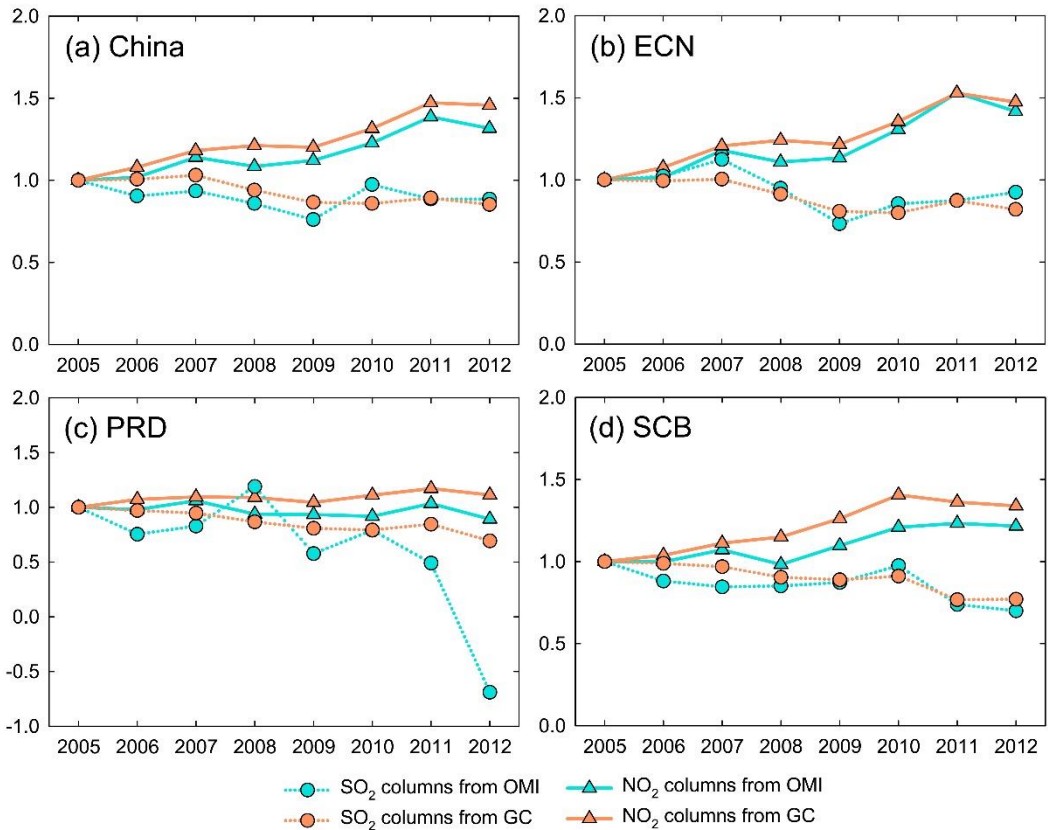

**Figure 2. The inter-annual trends of the simulated and observed tropospheric column densities for $SO_2$ and $NO_2$ over (a) China, (b) ECN, (c) PRD and (d) SCB during 2005–2012. All data are normalized by the value in 2005. The dotted lines with circles denote $SO_2$, and the solid lines with triangles denote $NO_2$. Cyan and orange represent the satellite (OMI) and model (GC) values, respectively.**

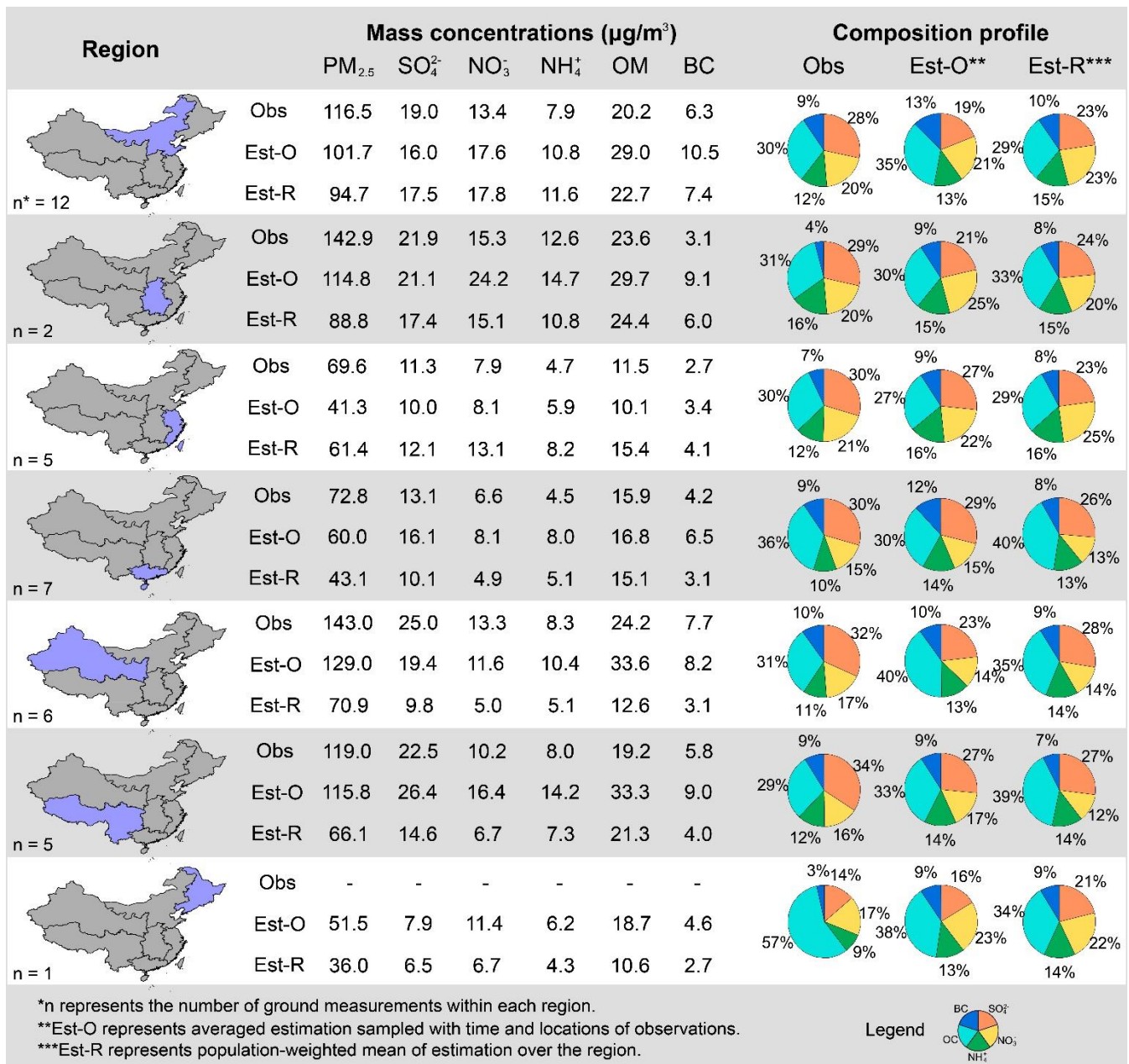

**Figure 3. PM₂.₅ composition concentrations and their fractions in seven regions over China. Region specific data were averaged from available ground measurements and satellite-derived data. Species shown only include sulfate, nitrate, ammonium, OM and BC. The observation data in Northeast China (Harbin) is only available in fractions, which is adopted from Wang et al. (2017). More details about the observation data and references are provided in Support Information.**

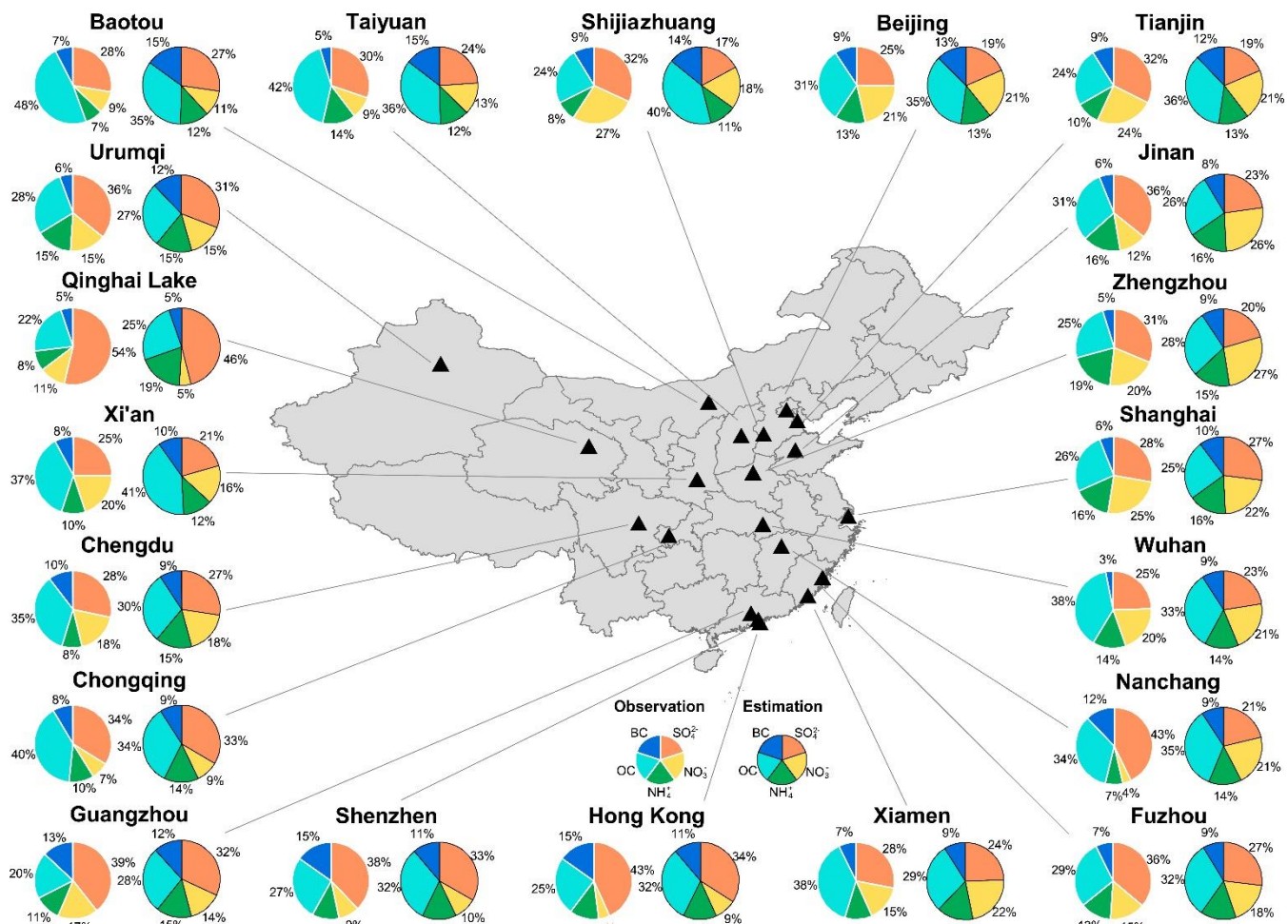

**Figure 4.** Observed and estimated PM$_{2.5}$ speciation for 20 major cities across China. Species shown only include sulfate, nitrate, ammonium, OC and BC. All observation data covered more than one year period. Details of the observations and references are provided in Support Information.

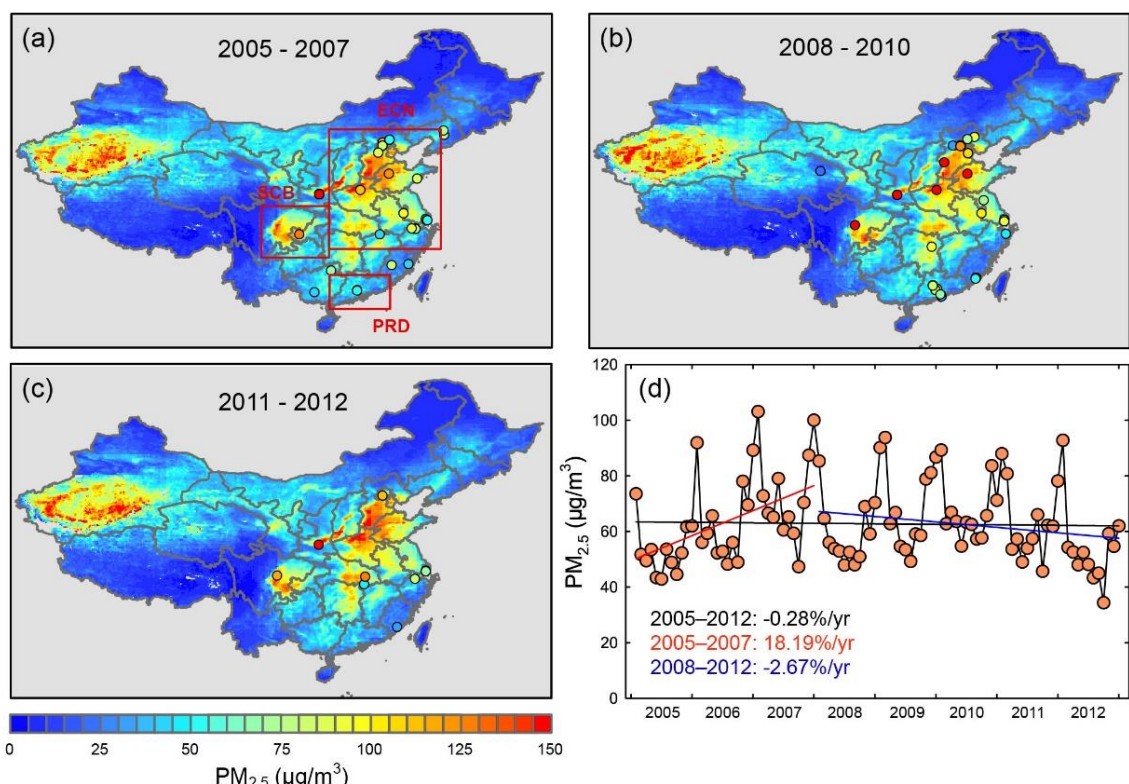

**Figure 5. Spatial distributions of averaged PM₂.₅ over China during (a) 2005–2007, (b) 2008–2010 and (c) 2011–2012 overlaid with ground measurements collected during the corresponding time period. (d) The population-weighted monthly mean PM₂.₅ concentrations over China and their regression trend. Boxed areas outline the regions defined in this study.**

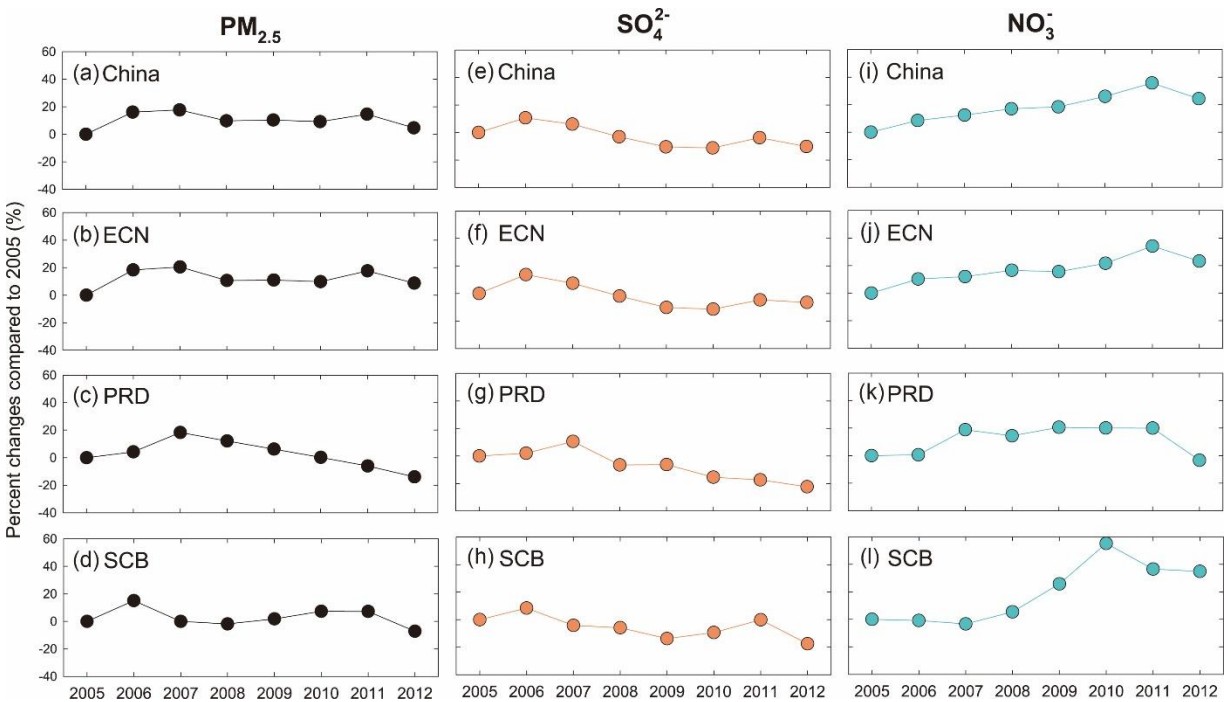

**Figure 6. Inter-annual percent changes of (a-d) PM$_{2.5}$, (e-h) SO$_4^{2-}$ and (i-l) NO$_3^-$ concentrations compared to 2005 over China, ECN, PRD and SCB.**

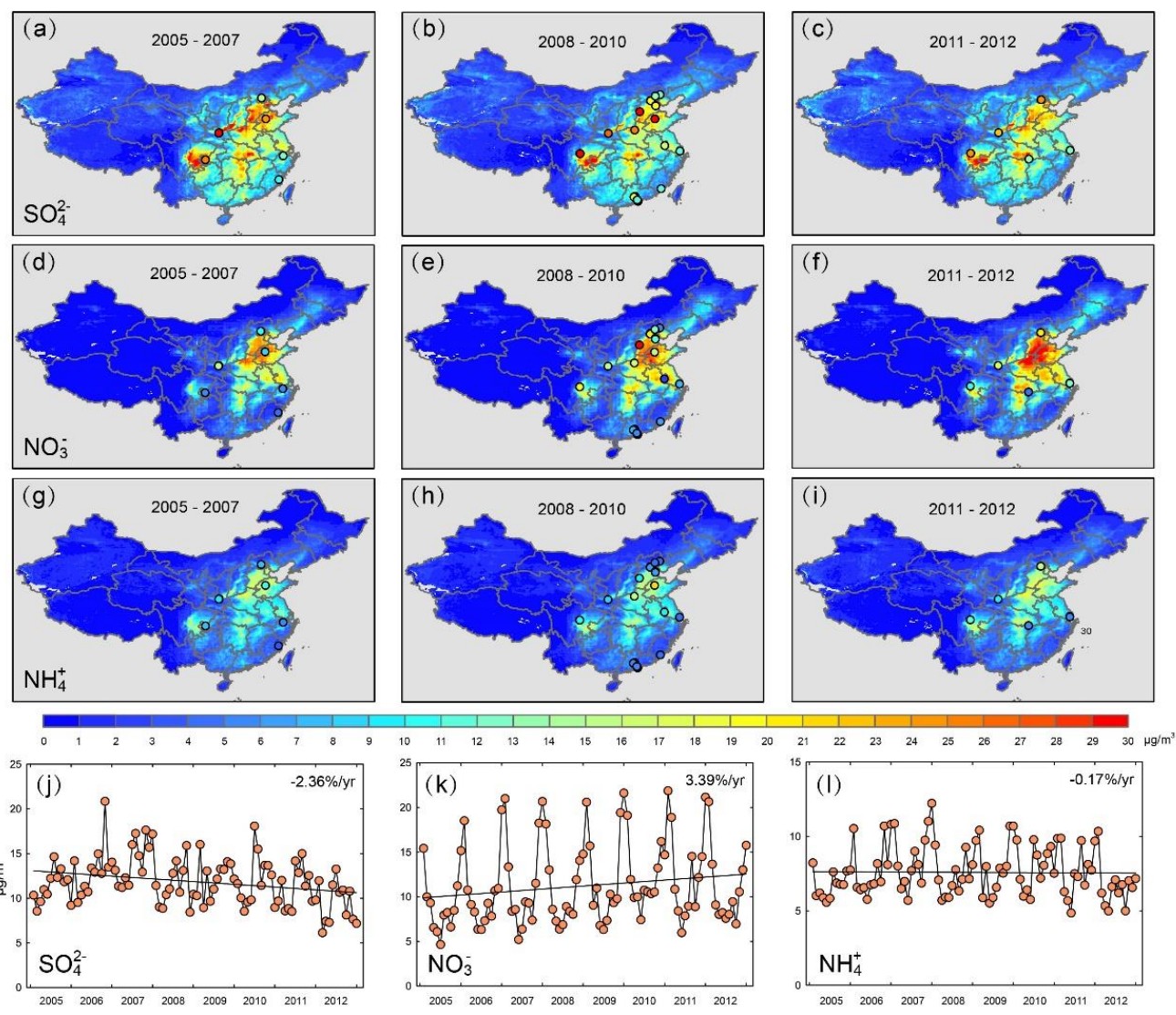

**Figure 7. Spatial distributions of averaged (a-c) $SO_4^{2-}$, (d-f) $NO_3^-$ and (g-i) $NH_4^+$ over China during 2005–2007, 2008–2010 and 2011–2012 overlaid with ground measurements collected during the corresponding time period. (j-l) The population-weighted monthly mean $SO_4^{2-}$, $NO_3^-$ and $NH_4^+$ concentrations over China and their regression trends.**

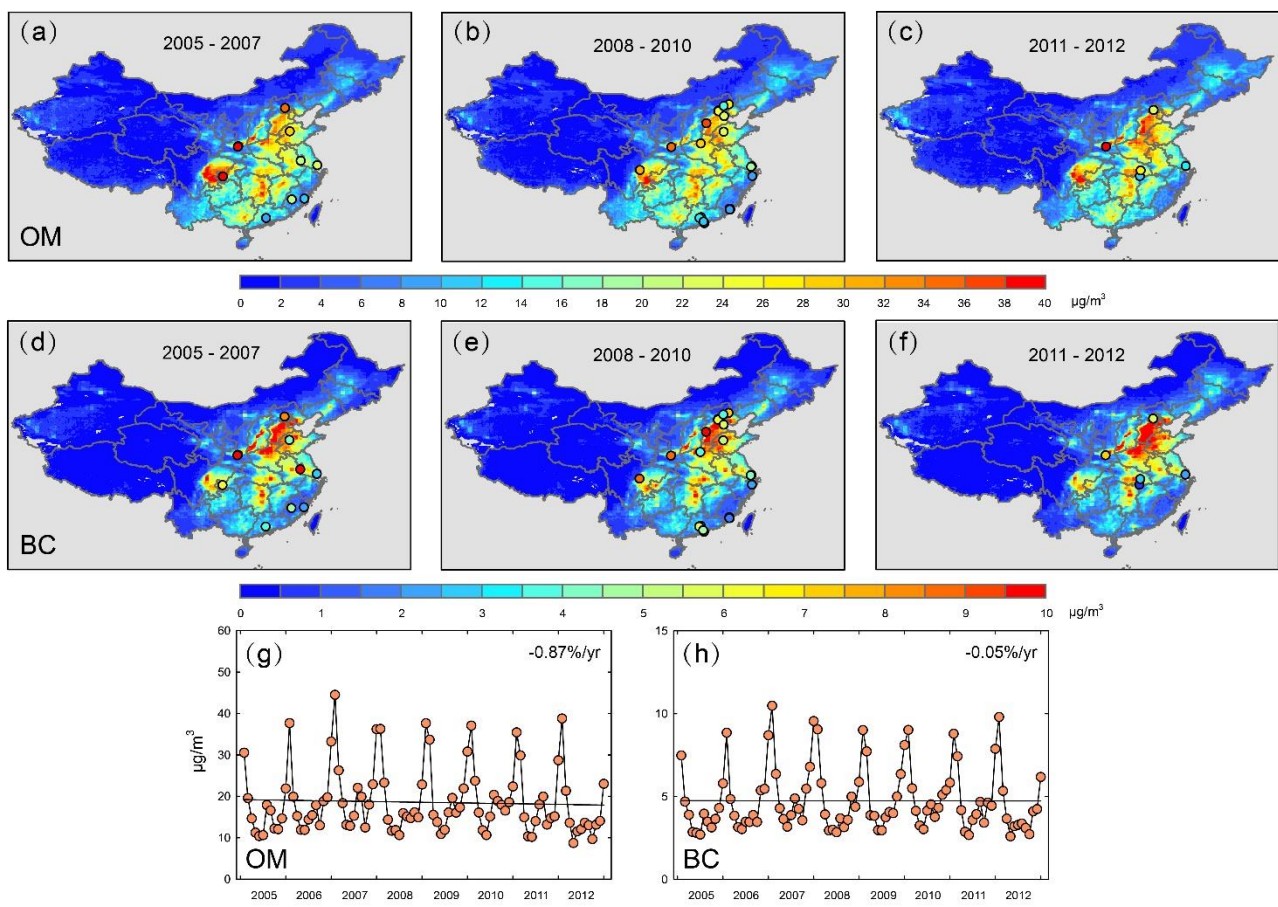

**Figure 8. Spatial distributions of averaged (a-c) OM and (d-f) BC over China during 2005–2007, 2008–2010 and 2011–2012 overlaid with ground measurements collected during the corresponding time period. (g-h) The population-weighted monthly mean OM and BC concentrations over China and their regression trends.**

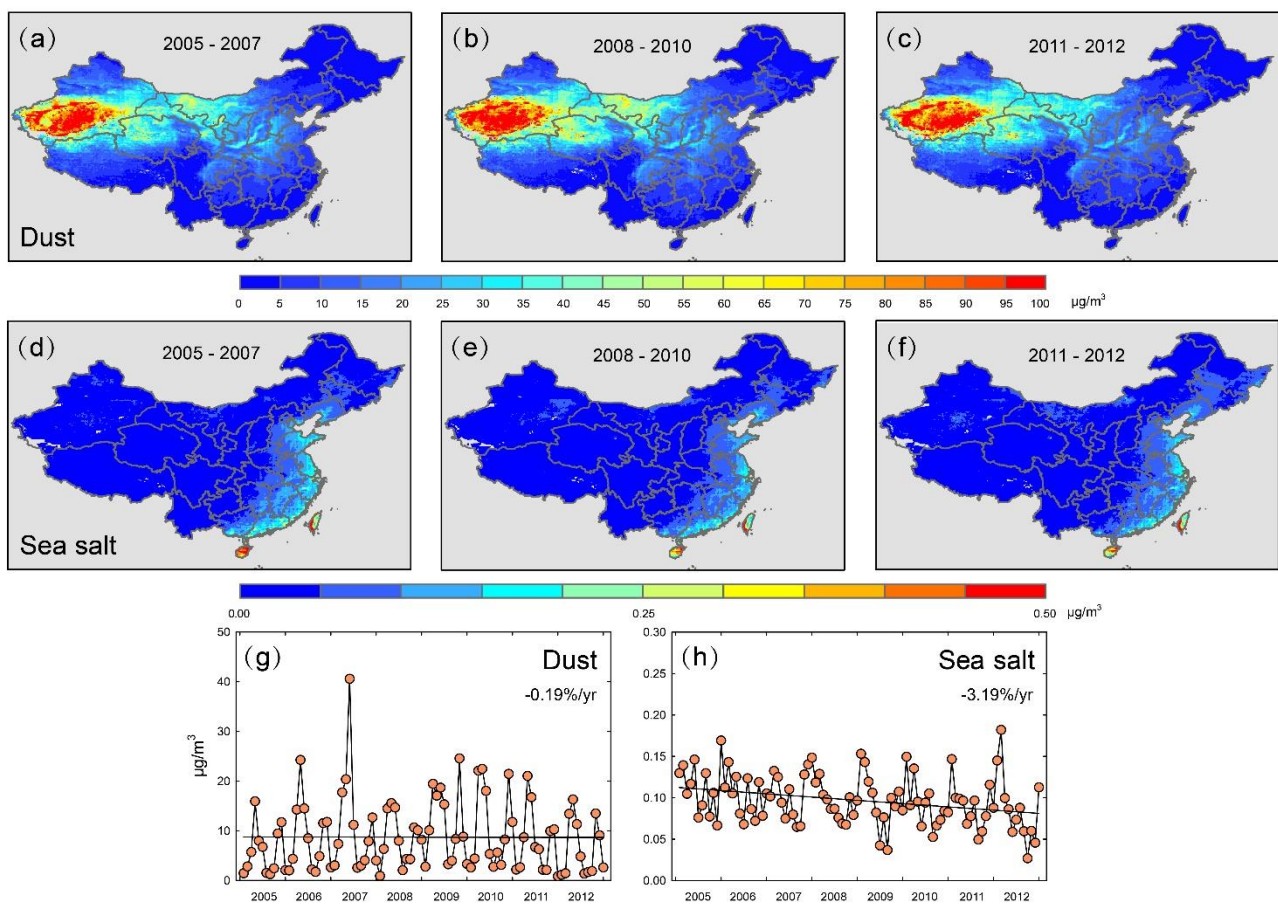

**Figure 9. Spatial distributions of averaged (a-c) dust and (d-f) sea salt over China during 2005–2007, 2008–2010 and 2011–2012. (g-h) The population-weighted monthly mean dust and sea salt concentrations over China and their regression trends.**

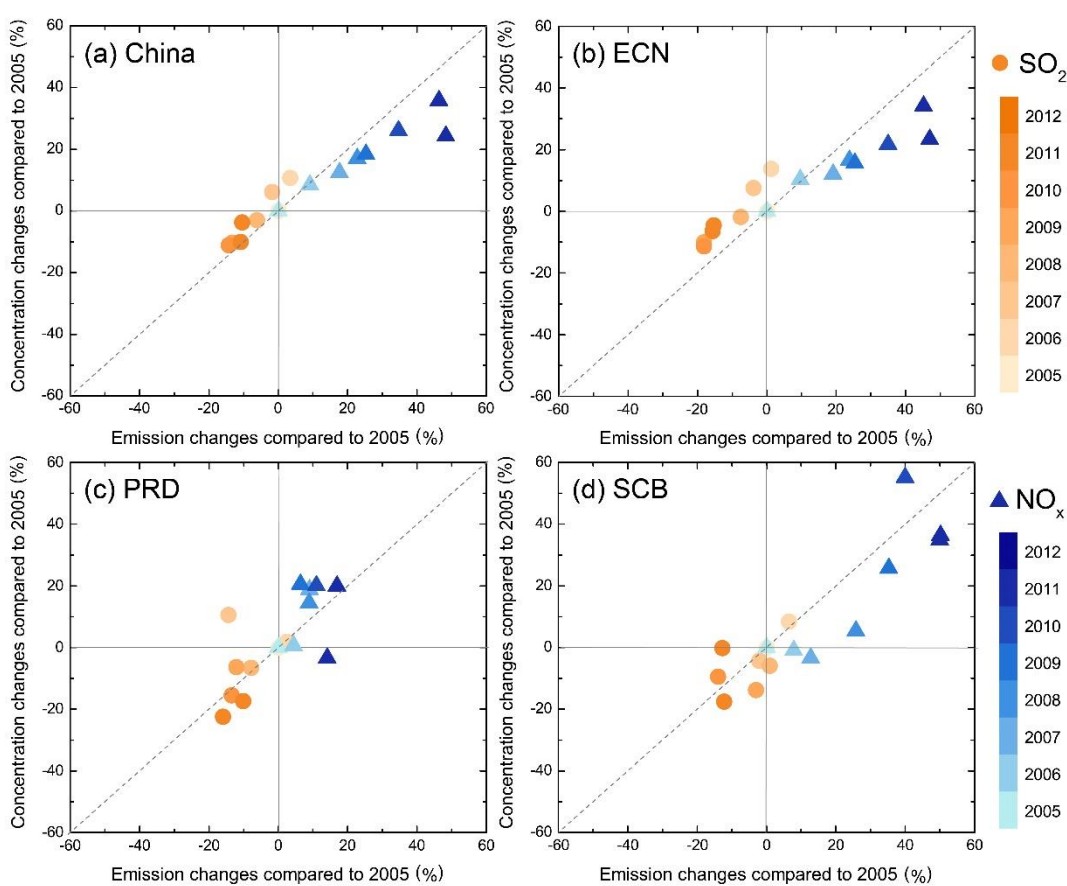

**Figure 10.** Comparisons of changes in the $SO_4^{2-}$ and $NO_3^-$ mass concentrations and their precursor emissions during 2005–2012. All data are presented as percent changes relative to 2005. The oranges circles and blue triangles represent $SO_4^{2-}$ and $NO_3^-$, respectively, and the shades of the symbols' colors denote the year.

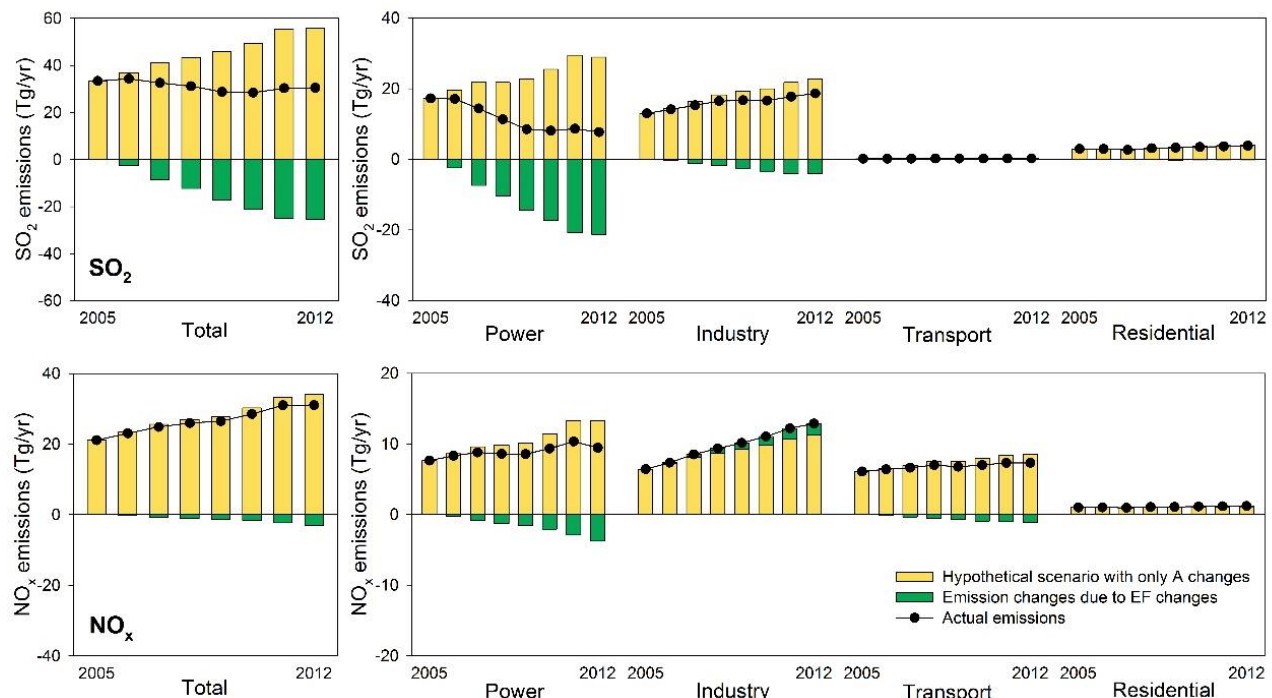

**Figure 11. SO₂ and NOₓ emissions by sector during 2005–2012 over China. The black circles denote the actual inter-annual emission trends estimated using the MEIC inventory. The yellow bars represent the hypothetical scenario involving only activity (A) changes. The green bars indicate the emission changes resulting from variations in the emission factor (EF). The top row illustrates the SO₂ emissions, and the bottom row presents the NOₓ emissions.**

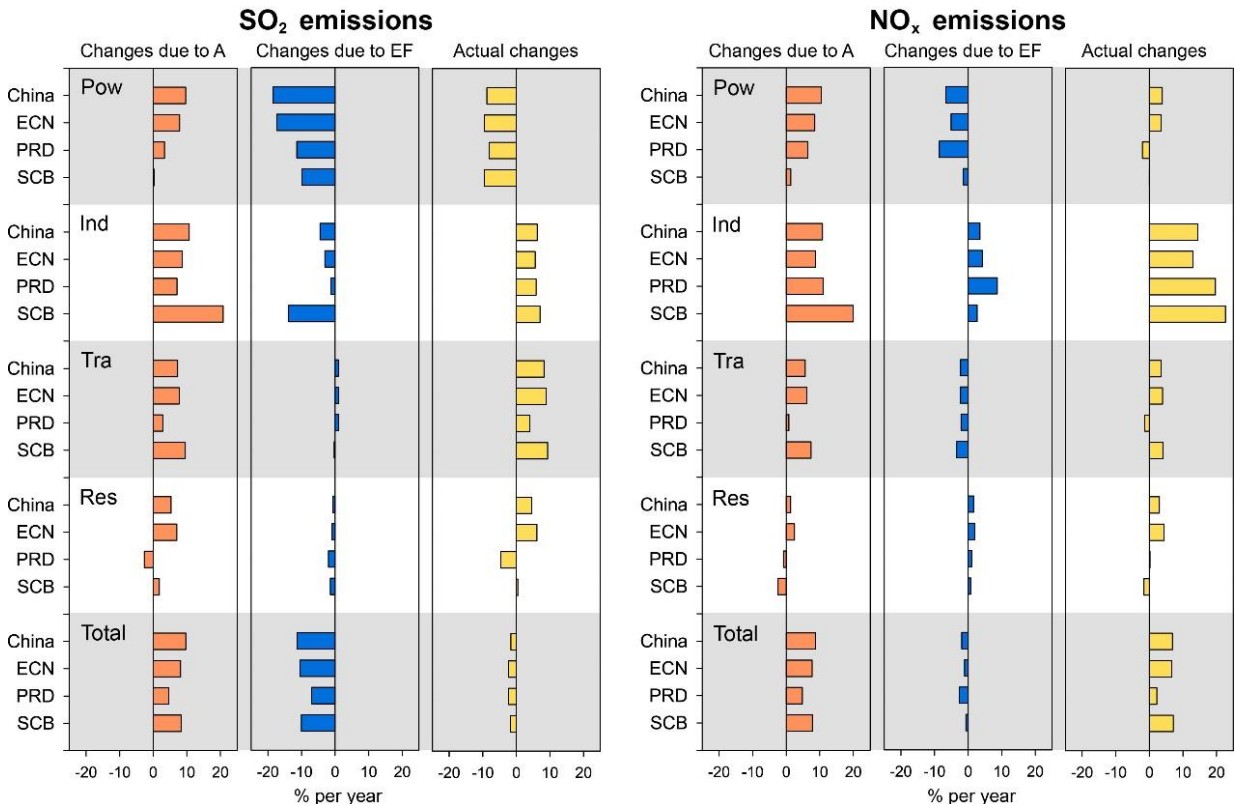

**Figure 12. Summary of the average emission changes (% per year) during 2005–2012 for the power (Pow), industry (Ind), transport (Tra) and residential (Res) sectors over China and the three regions. Orange bars indicate the emission changes caused by activity (A) changes, and blue bars represent the emission changes caused by emission factor (EF) changes. Yellow bars present the actual emission changes estimated using the MEIC model.**