# Peer review of "Chemical composition of ambient $PM_{2.5}$ over China and relationship to precursor emissions during 2005-2012"

_Atmospheric Chemistry and Physics, 2016_

## Referee Comment (RC1) · Anonymous Referee #1 · 12 Mar 2017

I think it is an important work adding to the discussion of the PM pollution in China. Indeed understanding of the past trends and chemical speciation is an important element of the work towards future strategy or evaluation of that strategy to reduce the PM exposure in China. Below are a number of comments and few questions that I hope are helpful to the authors and could potentially increase the value of the paper for several readers.

There is a lot of focus on high concentration areas and on the typical three regions for China. This is pretty common in many papers and there are good reasons for that. However, as Figure 1 shows, there are several stations in the range of 50-100 ug/m3 for which the model seem to underestimate the concentrations and these are in

the 'other' areas where possibly large population lives to and so for the overall health impact assessment it might be very important. I think this deserves more discussion in the text and possibly this is something that can be highlighted for further work.

The paper does not address the possible issues associated with changing aerosol load on satellite retrievals affecting potentially the trend analysis as well as the absolute comparison. The SI mentions issue of 'anomalies' in post 2007 period where less valid readings from OMI exist. I think this deserves a mention in the main paper and a short discussion of what it means for the error over time. I think some of that discussion can be added to the section 3.2.

Line 58-64: I think these two sentences should be reformulated. I do not believe that 'disparities in pollution characteristics' are the reason. It is the nature of air pollution that it does not know the borders and a mix of substances forms, travels over long distances making development of a comprehensive regional or national air quality strategy difficult, requiring knowledge of many different elements and measurements are essential factor. But only in combination with models (both CTM and remote sensing data) full understanding of the close and far sources on specific location can be understood and consequently managed.

Line 68-69: This sentence (conclusion) follows on the previous statements which to my mind do not fully characterize the complexity of the problem or the existing approaches. One has to consider that a lot of tools (CTMs) used in China were originally developed elsewhere where situation was different in many respects, including different level of pollutant concentrations, often existing networks of monitors with long time series, staff and laboratories with long term experience, existence of agencies monitoring pollution sources, etc. Obviously taking a set of tools from that context and trying to apply to China, or several Asian regions for that matter, will face challenges and the authors name a few. This sentence (line 68-69) reads a bit like the 'other' methods are disqualified to provide insight; I'd suggest to think of a more modest statement highlighting the additional approaches and analysis that can help to alleviate the issues and improve

understanding the pm2.5 problem in china in the past years.

More specific comments:

TITLE

Suggest replacing 'Chemical compositions' with 'Chemical composition'

ABSTRACT

Line 17: 'other correction factors'. . .I am not sure this is the best formulation; this is not informative. Suggest to reformulate

Line 23: Suggest replacing 'dominated' with 'dominating'

INTRODUCTION

Line 36: replace 'a mixture of complex materials from . . .' with 'a complex mixture originating from ..'

Line 41-43: It is not only 'direct' effects that are relevant and when mentioning BC, I would also add the reference to the 'Bounding BC study' by Bond et al. (2013) published in JGR. Consider rewriting the sentence; for example: 'Several components (e.g., sulfate, BC, OC) have significant impact on the global energy budget and consequently contribute to climate change (IPCC, 2013; Bond et al., 2013).

Line 43-44: This sentence shall be reformulated and I would focus on stressing on the role pm2,5 plays in haze formation as well as affecting visibility rather than public attention since currently also health issues attract attention. If you want to add a historical perspective then it should be a more comprehensive.

Line 54-55: 'cloud help design future plans' should be changed to 'could help design future control policies'

Line 56: add 'concentrations' after pm2.5

Line 95: Suggest replacing 'dominated' with 'dominating'

Line 97: suggest deleting 'figure out'

METHODOLOGY

Line 107: 'randomly distributed over time...'? A rather strange statement and I do not understand what the authors are trying to communicate here

Line 142: maybe modify to "...due to missing AOD data" Figure 1 and S1: I think it would be useful to see how the Figure S1 looks like when the points are shown with the same colour scale as in Figure 1 where regional allocation is indicated. From the Figure 1 it appears that there is pretty consistent about factor 2 underestimation for the low to moderate (well, in fact an average of 50-100 ug is not moderate but it is relative to 100-200 yg) concentrations in 'other' regions (yellow dots). I think this deserves few words of discussion in the text as there might be significant number of people leaving in these areas.

RESULTS

Line 259: Many readers might be interested about the possibly reasons for the overestimation of BC. Since this is a non-reactive species, does it mean that the emission are overestimated in bottom-up inventories or there are issues with the transport-deposition in the model?

Line 271: The first few lines of this section highlight the findings but are these really so new? I think a number of other papers have shown similar trends in emissions so this work compares the consistency of those estimates, at least in terms of emission trends. A couple of references can be added.

Line 278: Only vehicles are the cause? The level of control of SO2 and NOx has been different and only recently the power sector is asked to mitigate NOx while SO2 was longer on the agenda.

Line 280: Is S content the only reason? I thought that there has been also different requirements with respect to the emission standards for power and industrial sources

across provinces with Western Provinces having slower pace in introducing strict standards.

Line 359: This statement reads like it would be a fact but in fact it is an estimate and even if the total value appear to fit the overall satellite trends there are several uncertainties. I think that here and in other discussion in this section one needs to stress that these are estimates and also that a real confirmation might come from the CMS (cont monitoring systems) if such data will be available.

Line 362: It might be useful to highlight in this section of the paper how policies outside of the three focus regions affected the total emissions in China. TO give an example; Figure 10 shows for example for SO2 a larger change due to EF (power and total) in China as in any other three focus regions so I believe the contribution has to come from elsewhere.

Line 387: Linking to one of my earlier comments; this section could include also a word about the potential impact of changing aerosol load on the satellite retrievals

CONCLUSIONS

In general some repletion here of the discussion in chapter 4 so it could be shortened a bit. However, the issues I mentioned in the beginning of the review about the regions outside the three focus regions could be highlighted here as a possible area of further work.

---

## Referee Comment (RC2) · Anonymous Referee #2 · 26 Mar 2017

1. Page 1, line 19. Which population was used to do the population weighting? 2. Page 3, "2.2 Satellite-derived PM2.5 and chemical composition concentrations": what's spatial resolution for data integration and model fitting? what's the detailed meanings of AODCTM in equation (1) and (2)? Although the method has been published in Geng. et al. 2015, it would be easier for readers' understanding if more information given in this manuscript. 3. Why is population-weighted concentrations used to evaluate the inter-annual variation? What's the advantages of population-weighted concentrations comparing with unweighted concentrations in analysis the effects of controlling policies and emissions on PM2.5?

---

## Author Comment (AC2) · 7 May 2017

**Anonymous Referee #2:**

1. Page 1, line 19. Which population was used to do the population weighting?

**Response:** The population data used in this study are taken from the LandScan Global Population database (ORNL, 2010, Bright et al., 2011). We have added this information and the following reference to the revised manuscript.

Page 6, line 15 is revised as: 'The population-weighted mean mass concentrations of $PM_{2.5}$ over China were estimated using population data taken from the LandScan Global Population database (ORNL, 2010, Bright et al., 2011). The annual population-weighted mean concentrations were in the range of 63.9–75.2 $\mu g/m^3$ during 2005–2012 and were nearly twice the Chinese national standard.'

Reference: Bright, E. A., Coleman, P. R., Rose, A. N., and Urban, M. L.: LandScan 2010, in, 2010 ed., Oak Ridge National Laboratory, Oak Ridge, TN, 2011.

2. Page 3, "2.2 Satellite-derived PM2.5 and chemical composition concentrations": what's spatial resolution for data integration and model fitting? what's the detailed meanings of AODCTM in equation (1) and (2)? Although the method has been published in Geng. et al. 2015, it would be easier for readers' understanding if more information given in this manuscript.

**Response:** The satellite AOD data is at the spatial resolution of $0.1° \times 0.1°$, and the conversion factors are taken from the nested-grid GEOS-Chem model, which has a spatial resolution of $0.5° \times 0.666°$. The output datasets ($PM_{2.5}$ composition data) are at $0.1° \times 0.1°$. $AOD_{CTM}$ means AOD data comes from a CTM model.

We have revised the manuscript to better describe our method: 'The satellite-derived $PM_{2.5}$ concentration datasets used in this work were adopted from Geng et al. (2015). These data were calculated using satellite AOD and conversion factors between AOD and $PM_{2.5}$ simulated by a CTM, and the spatial resolution of the dataset is $0.1° \times 0.1°$. Following Philips et al. (2014), the satellite-derived chemical compositions of $PM_{2.5}$ at $0.1° \times 0.1°$ were estimated by applying composition-specific conversion factors to satellite AOD. The equations used for the $PM_{2.5}$ and composition calculations are:

$$PM_{2.5,satellite} = AOD_{satellite} \cdot \frac{PM_{2.5,CTM}}{AOD_{CTM}} \tag{1}$$

$$Composition^k_{satellite} = AOD_{satellite} \cdot \frac{Composition^k_{CTM}}{AOD_{CTM}} \tag{2}$$

where subscript 'satellite' and 'CTM' represent data from satellite and model respectively; $k$ represents different chemical compositions, including $SO_4^{2-}$, $NO_3^-$, $NH_4^+$, BC, OM, dust and sea salt, in this study.'

3. Why is population-weighted concentrations used to evaluate the inter-annual variation? What's the advantages of population-weighted concentrations comparing with unweighted concentrations in analysis the effects of controlling policies and emissions on PM2.5?

**Response:** We believe that the population-weighted mean concentrations can better reflect the changes of anthropogenic emissions, because anthropogenic emissions are usually emitted in populous regions. Putting more weights in populated area could partially avoid the influence of natural sources, such as dust from the desert. The northwestern part of China has very high $PM_{2.5}$ concentrations due to dust, but there is little population and emissions in that region. Using population-weighted mean concentrations can reduce the contribution of dusty region in the mean $PM_{2.5}$.

---

## Author Response (AR1)

**Anonymous Referee #1:**

I think it is an important work adding to the discussion of the PM pollution in China. Indeed understanding of the past trends and chemical speciation is an important element of the work towards future strategy or evaluation of that strategy to reduce the PM exposure in China. Below are a number of comments and few questions that I hope are helpful to the authors and could potentially increase the value of the paper for several readers.

**Response:** We thank Referee #1 for the encouragement and for the valuable comments to improve our manuscript. Responses to each point are addressed as below.

There is a lot of focus on high concentration areas and on the typical three regions for China. This is pretty common in many papers and there are good reasons for that. However, as Figure 1 shows, there are several stations in the range of 50-100 ug/m3 for which the model seem to underestimate the concentrations and these are in the 'other' areas where possibly large population lives to and so for the overall health impact assessment it might be very important. I think this deserves more discussion in the text and possibly this is something that can be highlighted for further work.

**Response:** We checked the locations of those estimates that were underestimated in 'other region'. They are located in Liaoning, Inner Mongolia and Fujian province, which are all places less populated than the typical three regions. But still, the underestimation could introduce biases in the health impact studies.

We have added some discussions in the revised manuscript: 'However, satellite-derived $PM_{2.5}$ concentrations were underestimated by a factor of 2 in some places outside the three regions (e.g. Liaoning, Inner Mongolia and Fujian province), which might affect the health impact estimates in these regions. The underestimation was mainly caused by the biases in modeled $PM_{2.5}$ concentrations, and the satellite-derived data improved compared to the model results (Figure S1).'

We have also added the analysis of $PM_{2.5}$ composition in seven regions and 20 major cities over China, and compared them with available ground measurements data, which is shown in the new Figure 3 and Figure 4.

In the conclusion part, we also added the following paragraph: 'In this study, our analysis mainly focused on the three selected regions (i.e., ECN, SCB and PRD), because they have high $PM_{2.5}$ levels, large anthropogenic emissions and large population densities. However, other places outside the three regions also experienced rapid industrialization and urbanization in recent years due to the national development strategies, which is important but has been paid less attention to. Future works could further improve the estimation of $PM_{2.5}$ composition

datasets in these area and investigate the spatiotemporal variations of $PM_{2.5}$ concentrations and its driving forces.'

The paper does not address the possible issues associated with changing aerosol load on satellite retrievals affecting potentially the trend analysis as well as the absolute comparison. The SI mentions issue of 'anomalies' in post 2007 period where less valid readings from OMI exist. I think this deserves a mention in the main paper and a short discussion of what it means for the error over time. I think some of that discussion can be added to the section 3.2.

**Response:** We thank the referee for raising up these issues. Aerosols can have a significant impact on the retrieval of tropospheric trace gases (e.g. $SO_2$ and $NO_2$) in their magnitude. For polluted regions like China, the annual mean $NO_2$ columns are enhanced by 15–40 % when considering aerosol effects (Lin et al., 2015). However, Boersma et al. (2004) showed that satellite-derived cloud fractions are also sensitive to aerosols with a high single scattering albedo. An increase in cloud fractions as a result of higher aerosol concentrations leads to a similar AMF correction for aerosols as would be accomplished through a direct radiative transfer calculation without cloud correction. So the trend of trace gases are less affected by the aerosol trend. The 'row anomaly' issue occurred in the OMI instrument since 2007 and affected the valid number of retrievals in both $SO_2$ and $NO_2$ vertical column densities. Different numbers of valid pixels among years could affect the trend of $SO_2$ and $NO_2$ column densities.
In this study, we used $SO_2$ and $NO_2$ vertical column densities to evaluate the model performance of simulating $PM_{2.5}$ precursors. We have added the following sentence to Section 3.2 to mention the uncertainties here: 'It is worth noting that the satellite retrieved $SO_2$ and $NO_2$ column densities have uncertainties in their trends because of the row anomaly issue happened to the CCD detectors in OMI instrument and the impact of changing aerosol loadings on the satellite retrievals, which might contribute to the discrepancies between modeled and satellite data.'

Line 58-64: I think these two sentences should be reformulated. I do not believe that 'disparities in pollution characteristics' are the reason. It is the nature of air pollution that it does not know the borders and a mix of substances forms, travels over long distances making development of a comprehensive regional or national air quality strategy difficult, requiring knowledge of many different elements and measurements are essential factor. But only in combination with models (both CTM and remote sensing data) full understanding of the close and far sources on specific location can be understood and consequently managed.

**Response:** We thank the referee for the valuable suggestion. We have reformulated the sentences as: 'Measurements from individual cities are insufficient to support a comprehensive national analysis or health impact studies because air quality issues are usually regional problems and require knowledge of many different elements. Full understanding of the

pollution sources can be achieved in combination with CTMs (Wang et al., 2013; Xing et al., 2015).'

Line 68-69: This sentence (conclusion) follows on the previous statements which to my mind do not fully characterize the complexity of the problem or the existing approaches. One has to consider that a lot of tools (CTMs) used in China were originally developed elsewhere where situation was different in many respects, including different level of pollutant concentrations, often existing networks of monitors with long time series, staff and laboratories with long term experience, existence of agencies monitoring pollution sources, etc. Obviously taking a set of tools from that context and trying to apply to China, or several Asian regions for that matter, will face challenges and the authors name a few. This sentence (line 68-69) reads a bit like the 'other' methods are disqualified to provide insight; I'd suggest to think of a more modest statement highlighting the additional approaches and analysis that can help to alleviate the issues and improve understanding the pm2.5 problem in china in the past years.

**Response:** We thank the referee for the constructive suggestions. We have revised the sentence to more accurately describe the limitations in CTMs as following: 'However, CTMs have limitations in $PM_{2.5}$ simulations over China, since many models have been originally developed in other regions that have different pollution levels compared to China. Application of these models in China might introduce problems including missing precursors and formation mechanisms of secondary organic aerosols (Baek et al., 2011) and the lack of heterogeneous reactions, which may lead to underestimations of sulfate in haze events (Wang et al., 2014; Zheng et al., 2015)'
We also reformulated the last sentence as: 'Therefore, additional information is needed to alleviate the issues and improve the simulations of historical $PM_{2.5}$ chemical compositions.'

More specific comments:

**TITLE**

Suggest replacing 'Chemical compositions' with 'Chemical composition'

**Response:** Revised.

**ABSTRACT**

Line 17: 'other correction factors'. . .I am not sure this is the best formulation; this is not informative. Suggest to reformulate

**Response:** We have revised the sentence as 'We estimated the changes in chemical composition of ambient $PM_{2.5}$ over China during 2005–2012 using satellite-based aerosol optical depth (AOD) data and the GEOS-Chem chemical transport model, and investigated the driving forces behind the changes by examining the changes in precursor emissions using a bottom-up emission inventory.'

Line 23: Suggest replacing 'dominated' with 'dominating'

**Response:** Revised.

**INTRODUCTION**

Line 36: replace 'a mixture of complex materials from . . .' with 'a complex mixture originating from ..'

**Response:** Revised.

Line 41-43: It is not only 'direct' effects that are relevant and when mentioning BC, I would also add the reference to the 'Bounding BC study' by Bond et al. (2013) published in JGR. Consider rewriting the sentence; for example: 'Several components (e.g., sulfate, BC, OC) have significant impact on the global energy budget and consequently contribute to climate change (IPCC, 2013; Bond et al., 2013).

**Response:** Revised as suggested: 'Some components (e.g., sulfate, OC and BC) have significant impacts on the global energy budget system and consequently contribute to climate change (Bond et al., 2013; IPCC, 2013).'

Line 43-44: This sentence shall be reformulated and I would focus on stressing on the role pm2,5 plays in haze formation as well as affecting visibility rather than public attention since currently also health issues attract attention. If you want to add a historical perspective then it should be a more comprehensive.

**Response:** Revised as suggested: '$PM_{2.5}$ can also trigger visibility degradation or cause extreme haze events.'

Line 54-55: 'cloud help design future plans' should be changed to 'could help design future control policies'

**Response:** Revised.

Line 56: add 'concentrations' after pm2.5

**Response:** Revised.

Line 95: Suggest replacing 'dominated' with 'dominating'

**Response:** Revised.

Line 97: suggest deleting 'figure out'

**Response:** Revised.

**METHODOLOGY**

Line 107: 'randomly distributed over time. . .'? A rather strange statement and I do not understand what the authors are trying to communicate here

**Response:** In an ideal condition, the measurements data used to evaluate the estimations should cover the whole study time period from 2005 to 2012. However, we don't have temporal continuous observations in China prior to 2013, and the measurements data used in this study are collected from publications. These observation data only represent parts of the study period (e.g., several months or years), however, they were randomly distributed during 2005-2012, which we believe can be representative for the study time period. To better describe the issue, we have revised the sentence as: 'Although spatio-temporal continuous observation data are unavailable, the collected measurements cover most of the eastern provinces, are randomly distributed in time, and are considered to be representative of our study time and region.'

Line 142: maybe modify to ". . .due to missing AOD data"

**Response:** Revised.

Figure 1 and S1: I think it would be useful to see how the Figure S1 looks like when the points are shown with the same colour scale as in Figure 1 where regional allocation is indicated. From the Figure 1 it appears that there is pretty consistent about factor 2 underestimation for the low to moderate (well, in fact an average of 50-100 ug is not moderate but it is relative to 100-200 yg) concentrations in 'other' regions (yellow dots). I think this deserves few words of discussion in the text as there might be significant number of people leaving in these areas.

**Response:** We have revised Figure S1 to show same colour scale as Figure 1.

We also checked the locations of those estimates that were about factor 2 underestimation in 'other region'. They are located in Liaoning, Inner Mongolia and Fujian province, which are all places less populated than the typical three regions. But still, the underestimation could introduce biases in the health impact studies. We have added some discussions here: 'However, satellite-derived $PM_{2.5}$ concentrations were underestimated by a factor of 2 in some places outside the three regions (e.g. Liaoning, Inner Mongolia and Fujian province), which might affect the health impact estimates in these regions. The underestimation was mainly caused by the biases in modeled $PM_{2.5}$ concentrations, and the satellite-derived data improved compared to the model results (Figure S1).'

**RESULTS**

Line 259: Many readers might be interested about the possibly reasons for the overestimation of BC. Since this is a non-reactive species, does it mean that the emission are overestimated in bottom-up inventories or there are issues with the transport-deposition in the model?

**Response:** The GEOS-Chem modeled BC concentrations are underestimated compared to ground measurement data as shown in Figure S1, which is consistent with previous studies (Zhang et al., 2015). This underestimation is possibly due to underestimation in BC emission inventory (Zhang et al., 2015).

The satellite-derived BC concentrations in this study are overestimated compared to observations (Figure 1), which is caused by the overestimation of simulated BC fractions in $PM_{2.5}$. As can be seen in Figure S1, GEOS-Chem modeled BC is less underestimated than other species like sulfate and OM, which resulted in higher BC fractions in the simulated $PM_{2.5}$.

We have revised the sentence as: 'Satellite-based data performed well for estimating OM concentrations but tended to overestimate the BC concentrations in polluted regions, which was caused by the overestimation of modeled BC fractions by the GEOS-Chem model'

Line 271: The first few lines of this section highlight the findings but are these really so new? I think a number of other papers have shown similar trends in emissions so this work compares the consistency of those estimates, at least in terms of emission trends. A couple of references can be added.

**Response:** We have added reference that describe the SNA concentrations trend using CTMs: 'Based on the analysis described above, we found that sulfate and nitrate were the two dominant components driving the variations in the $PM_{2.5}$ concentration during 2005–2012, which is consistent with previous studies using CTMs (Xing et al., 2015).'

We also added reference in Section 4.2 when describing $SO_2$ and $NO_x$ emissions: 'The emissions trends estimated by the MEIC model are consistent with other studies (Lu et al., 2010; Lu et al., 2011; Zhao et al., 2013; Zhao et al., 2013).'

Line 278: Only vehicles are the cause? The level of control of SO2 and NOx has been different and only recently the power sector is asked to mitigate NOx while SO2 was longer on the agenda.

**Response:** We have revised the discussion here as: 'The $NO_3^-/SO_4^{2-}$ ratios were larger in the eastern part of China, because eastern part of China had stricter emissions standards and higher vehicle populations (Zheng et al., 2014), and in western China (e.g., Sichuan, Chongqing, and Ningxia), where coal with higher sulfur contents is burned (Tang et al., 2008), resulting in higher emission factors of $SO_2$. During 2005–2012, the $NO_3^-/SO_4^{2-}$ ratios over China exhibited an increasing trend, further supporting the changes in the relative abundances of sulfate and nitrate in the atmosphere and the distinct process of controlling $SO_2$ and $NO_x$ emissions in China.'

Line 280: Is S content the only reason? I thought that there has been also different requirements with respect to the emission standards for power and industrial sources across provinces with Western Provinces having slower pace in introducing strict standards.

**Response:** We have revised the discussion here as: 'The $NO_3^-/SO_4^{2-}$ ratios were larger in the eastern part of China, because eastern part of China had stricter emissions standards and higher vehicle populations (Zheng et al., 2014), and in western China (e.g., Sichuan, Chongqing, and Ningxia), where coal with higher sulfur contents is burned (Tang et al., 2008), resulting in higher emission factors of $SO_2$. During 2005–2012, the $NO_3^-/SO_4^{2-}$ ratios over China exhibited an increasing trend, further supporting the changes in the relative abundances of sulfate and nitrate in the atmosphere and the distinct process of controlling $SO_2$ and $NO_x$ emissions in China.'

Line 359: This statement reads like it would be a fact but in fact it is an estimate and even if the total value appear to fit the overall satellite trends there are several uncertainties. I think that here and in other discussion in this section one needs to stress that these are estimates and also that a real confirmation might come from the CMS (cont monitoring systems) if such data will be available.

**Response:** We thank the referee for pointing out the issue. The $PM_{2.5}$ composition trend and precursor emission trend are estimates of this study and have their uncertainties. We have revised our manuscript as suggested and also provide references here to support our conclusion:

'In comparison to 2011, the $NO_x$ emissions reduction caused by a decline in the activity level in the power sector in 2012 was estimated to be 26.5 Gg, much less than the estimated emission reduction of 897.5 Gg, implying that SCR promotion might have begun to take effect. The reduction of $NO_x$ emissions have also been noticed by other studies using satellite retrievals (de Foy et al., 2016; Liu et al., 2016; van Der A et al., 2016).'

Line 362: It might be useful to highlight in this section of the paper how policies outside of the three focus regions affected the total emissions in China. TO give an example; Figure 10 shows for example for SO2 a larger change due to EF (power and total) in China as in any other three focus regions so I believe the contribution has to come from elsewhere.

**Response:** Regions outside the three selected regions had larger reduction rates due to EF in power sector, which contributed to the larger change due to EF over China. This is related to the study period selected in this region, i.e. 2005~2012. In the year 2005, more developed regions like ECN, PRD has already took actions to reduce the power emissions, which had a higher $SO_2$ control efficiency above the national average level. At meanwhile, $SO_2$ control efficiency in other regions were below the national average level.

In the year 2012, FGD devices were installed in most power plants over China. The $SO_2$ control efficiency are similar among different regions. Therefore, the change between 2012 and 2005 were larger in 'other region' that the three selected regions.

Line 387: Linking to one of my earlier comments; this section could include also a word about the potential impact of changing aerosol load on the satellite retrievals

**Response:** Aerosols can have a significant impact on the retrieval of tropospheric trace gases (e.g. $SO_2$ and $NO_2$). For polluted regions like China, the annual mean $NO_2$ columns are enhanced by 15–40 % when considering aerosol effects (Lin et al., 2015). However, Boersma et al. (2004) showed that satellite-derived cloud fractions are also sensitive to aerosols with a high single scattering albedo. An increase in cloud fractions as a result of higher aerosol concentrations leads to a similar AMF correction for aerosols as would be accomplished through a direct radiative transfer calculation without cloud correction. So the trend of trace gases are less affected.

However, we used $SO_2$ and $NO_2$ vertical column densities here to evaluate the model performance of simulating $PM_{2.5}$ precursors. We have added the following sentence to Section 3.2 to mention the uncertainties here: 'The satellite retrieved $SO_2$ and $NO_2$ column densities have uncertainties in their trends because of the row anomaly issue happened to the CCD detectors in OMI instrument and the impact of changing aerosol loadings on the satellite retrievals.'

**CONCLUSIONS**

In general some repletion here of the discussion in chapter 4 so it could be shortened a bit. However, the issues I mentioned in the beginning of the review about the regions outside the three focus regions could be highlighted here as a possible area of further work.

**Response:** Thanks for the suggestions. We have deleted the first paragraph in conclusion to avoid repletion, and added the following paragraph to the revised manuscript: 'In this study, our analysis mainly focused on the three selected regions (i.e., ECN, SCB and PRD), because they have high $PM_{2.5}$ levels, large anthropogenic emissions and large population densities. However, other places outside the three regions also experienced rapid industrialization and urbanization in recent years due to the national development strategies, which is important but has been paid less attention to. Future works could further improve the estimation of $PM_{2.5}$ composition datasets in these area and investigate the spatiotemporal variations of $PM_{2.5}$ concentrations.'

**Response:** The satellite AOD data is at the spatial resolution of $0.1° \times 0.1°$, and the conversion factors are taken from the nested-grid GEOS-Chem model, which has a spatial resolution of $0.5° \times 0.666°$. The output datasets (PM$_{2.5}$ composition data) are at $0.1° \times 0.1°$. AOD$_{CTM}$ means AOD data comes from a CTM model.

We have revised the manuscript to better describe our method: 'The satellite-derived PM$_{2.5}$ concentration datasets used in this work were adopted from Geng et al. (2015). These data were calculated using satellite AOD and conversion factors between AOD and PM$_{2.5}$ simulated by a CTM, and the spatial resolution of the dataset is $0.1° \times 0.1°$. Following Philips et al. (2014), the satellite-derived chemical compositions of PM$_{2.5}$ at $0.1° \times 0.1°$ were estimated by applying composition-specific conversion factors to satellite AOD. The equations used for the PM$_{2.5}$ and composition calculations are:

$$PM_{2.5,satellite} = AOD_{satellite} \cdot \frac{PM_{2.5,CTM}}{AOD_{CTM}}$$
(1)

$$Composition^k_{satellite} = AOD_{satellite} \cdot \frac{Composition^k_{CTM}}{AOD_{CTM}}$$
(2)

where subscript 'satellite' and 'CTM' represent data from satellite and model respectively; $k$ represents different chemical compositions, including $SO_4^{2-}$, $NO_3^-$, $NH_4^+$, BC, OM, dust and sea salt, in this study.'

3. Why is population-weighted concentrations used to evaluate the inter-annual variation? What's the advantages of population-weighted concentrations comparing with unweighted concentrations in analysis the effects of controlling policies and emissions on PM2.5?

**Response:** We believe that the population-weighted mean concentrations can better reflect the changes of anthropogenic emissions, because anthropogenic emissions are usually emitted in populous regions. Putting more weights in populated area could partially avoid the influence of natural sources, such as dust from the desert. The northwestern part of China has very high PM$_{2.5}$ concentrations due to dust, but there is little population and emissions in that region. Using population-weighted mean concentrations can reduce the contribution of dusty region in the mean PM$_{2.5}$.

[revised manuscript text omitted]

*n represents the number of ground measurements within each region.
**Est-O represents averaged estimation sampled with time and locations of observations.
***Est-R represents population-weighted mean of estimation over the region.

**Figure 3. PM₂.₅ composition concentrations and their fractions in seven regions over China. Region specific data were averaged from available ground measurements and satellite-derived data. Species shown only include sulfate, nitrate, ammonium, OM and BC. The observation data in Northeast China (Harbin) is only available in fractions, which is adopted from Wang et al. (2017). More details about the observation data and references are provided in Support Information.**

[Figure]

**Figure 4. Observed and estimated PM$_{2.5}$ speciation for 20 major cities across China. Species shown only include sulfate, nitrate, ammonium, OC and BC. All observation data covered more than one year period. Details of the observations and references are provided in Support Information.**

[Figure]

**Figure 5. Spatial distributions of averaged PM$_{2.5}$ over China during (a) 2005–2007, (b) 2008–2010 and (c) 2011–2012 overlaid with ground measurements collected during the corresponding time period. (d) The population-weighted monthly mean PM$_{2.5}$ concentrations over China and their regression trend. Boxed areas outline the regions defined in this study.**

[Figure]

**Figure 6. Inter-annual percent changes of (a-d) PM$_{2.5}$, (e-h) SO$_4^{2-}$ and (i-l) NO$_3^-$ concentrations compared to 2005 over China, ECN, PRD and SCB.**

[Figure]

**Figure 7. Spatial distributions of averaged (a-c) $SO_4^{2-}$, (d-f) $NO_3^-$ and (g-i) $NH_4^+$ over China during 2005–2007, 2008–2010 and 2011–2012 overlaid with ground measurements collected during the corresponding time period. (j-l) The population-weighted monthly mean $SO_4^{2-}$, $NO_3^-$ and $NH_4^+$ concentrations over China and their regression trends.**

[Figure]

**Figure 8. Spatial distributions of averaged (a-c) OM and (d-f) BC over China during 2005–2007, 2008–2010 and 2011–2012 overlaid with ground measurements collected during the corresponding time period. (g-h) The population-weighted monthly mean OM and BC concentrations over China and their regression trends.**

[Figure]

**Figure 9. Spatial distributions of averaged (a-c) dust and (d-f) sea salt over China during 2005–2007, 2008–2010 and 2011–2012. (g-h) The population-weighted monthly mean dust and sea salt concentrations over China and their regression trends.**

[Figure]

**Figure 10. Comparisons of changes in the** $SO_4^{2-}$ **and** $NO_3^-$ **mass concentrations and their precursor emissions during 2005–2012. All data are presented as percent changes relative to 2005. The oranges circles and blue triangles represent** $SO_4^{2-}$ **and** $NO_3^-$, **respectively, and the shades of the symbols' colors denote the year.**

[Figure]

**Figure 11. SO₂ and NOₓ emissions by sector during 2005–2012 over China. The black circles denote the actual inter-annual emission trends estimated using the MEIC inventory. The yellow bars represent the hypothetical scenario involving only activity (A) changes. The green bars indicate the emission changes resulting from variations in the emission factor (EF). The top row illustrates the SO₂ emissions, and the bottom row presents the NOₓ emissions.**

[Figure]

**Figure 12. Summary of the average emission changes (% per year) during 2005–2012 for the power (Pow), industry (Ind), transport (Tra) and residential (Res) sectors over China and the three regions. Orange bars indicate the emission changes caused by activity (A) changes, and blue bars represent the emission changes caused by emission factor (EF) changes. Yellow bars present the actual emission changes estimated using the MEIC model.**